# Predictive biomarkers for survival benefit with ramucirumab in urothelial cancer in the RANGE trial

Michiel S. van der Heijden[1✉], Thomas Powles[2], Daniel Petrylak[3], Ronald de Wit[4], Andrea Necchi[5], Cora N. Sternberg[6], Nobuaki Matsubara[7], Hiroyuki Nishiyama[8], Daniel Castellano[9], Syed A. Hussain[10], Aristotelis Bamias[11], Georgios Gakis[12], Jae-Lyun Lee[13], Scott T. Tagawa[14], Ulka Vaishampayan[15], Jeanny B. Aragon-Ching[16], Bernie J. Eigl[17], Rebecca R. Hozak[18], Erik R. Rasmussen[19], Meng Summer Xia[18], Ryan Rhodes[18], Sameera Wijayawardana[18], Katherine M. Bell-McGuinn[18], Amit Aggarwal[18] & Alexandra Drakaki[20]

The RANGE study (NCT02426125) evaluated ramucirumab (an anti-VEGFR2 monoclonal antibody) in patients with platinum-refractory advanced urothelial carcinoma (UC). Here, we use programmed cell death-ligand 1 (PD-L1) immunohistochemistry (IHC) and transcriptome analysis to evaluate the association of immune and angiogenesis pathways, and molecular subtypes, with overall survival (OS) in UC. Higher PD-L1 IHC and immune pathway scores, but not angiogenesis scores, are associated with greater ramucirumab OS benefit. Additionally, Basal subtypes, which have higher PD-L1 IHC and immune/angiogenesis pathway scores, show greater ramucirumab OS benefit compared to Luminal subtypes, which have relatively lower scores. Multivariable analysis suggests patients from East Asia as having lower immune/angiogenesis signature scores, which correlates with decreased ramucirumab OS benefit. Our data highlight the utility of multiple biomarkers including PD-L1, molecular subtype, and immune phenotype in identifying patients with UC who might derive the greatest benefit from treatment with ramucirumab.

[1] The Netherlands Cancer Institute, Amsterdam, The Netherlands. [2] Barts Cancer Institute, Queen Mary University of London, London, UK. [3] Smilow Cancer Hospital at Yale New Haven, Yale New Haven Hospital, New Haven, CT, USA. [4] Erasmus MC Cancer Institute, Rotterdam, The Netherlands. [5] Vita-Salute San Raffaele University and IRCCS San Raffaele Hospital, Milan, Italy. [6] Caryl and Israel Englander Institute for Precision Medicine, Weill Cornell Medicine, Meyer Cancer Center, New York-Presbyterian Hospital, New York, NY, USA. [7] National Cancer Center Hospital East, Chiba, Japan. [8] University of Tsukuba, Tsukuba, Ibaraki, Japan. [9] Hospital Universitario 12 de Octubre, Madrid, Spain. [10] University of Sheffield, Department of Oncology and Metabolism, Sheffield, UK. [11] National and Kapodistrian University of Athens, Athens, Greece. [12] University Hospital of Würzburg, Würzburg, Germany. [13] Asan Medical Center, Urologic Cancer Center, Seoul, South Korea. [14] Weill Cornell Medical College, Department of Genitourinary Oncology, New York, NY, USA. [15] University of Michigan Ann Arbor, Ann Arbor, MI, USA. [16] Inova Schar Cancer Institute, Fairfax, VA, USA. [17] BC Cancer, Vancouver, BC, Canada. [18] Eli Lilly and Company, Indianapolis, IN, USA. [19] Eli Lilly and Company, New York, NY, USA. [20] David Geffen School of Medicine, Division of Hematology and Oncology, UCLA, Los Angeles, CA, USA. ✉email: ms.vd.heijden@nki.nl

dentification of biomarkers to predict which subset of patients will derive the most benefit from a particular therapy is needed to optimize targeted therapies. One emerging biomarker for urothelial carcinoma (UC) is expression of programmed cell death-ligand 1 (PD-L1) which, together with its receptor, programmed cell death protein (PD-1), are the targets of multiple immune-checkpoint inhibitors[1–7], which have been shown to improve overall survival (OS) in the second-line (KEYNOTE-45)[3] and switch maintenance settings (JAVELIN 100)[5]. However, PD-L1 expression did not always correlate with efficacy outcomes. Patients below the tumor PD-L1 expression cutoff could still respond to immune-checkpoint inhibitors, suggesting that use of single-gene biomarkers may not be representative of the complexity of UC[8,9].

Urothelial carcinoma has been comprehensively genomically characterized[10,11]. Along with single-protein biomarkers, this more complex profiling may provide another useful way to study biomarkers in UC and gain insight into disease biology and targets for novel therapies. Several UC molecular classification schemes using different genomic platforms and patient datasets have now been described[10–15]. These schemes are based on "basal-like" and "luminal-like" cellular phenotypes that are further refined into discrete subclassifications. The Cancer Genome Atlas (TCGA) identified five expression subtypes: Basal Squamous, Luminal, Luminal Infiltrated, Luminal Papillary, and Neuronal[15]. Seiler et al. developed a single-sample genomic subtyping classifier (GSC) to predict four consensus subtypes: Basal, (Basal) Claudin Low, Luminal, and Luminal Infiltrated[16]. Recently, a single-sample consensus molecular classifier derived by the Bladder Cancer Molecular Taxonomy Group has been proposed. This ConsensusMIBC (muscle-invasive bladder cancer) classifier groups tumor types into six subtypes: Basal/Squamous, Luminal Papillary, Luminal Non-Specified, Luminal Unstable, Stroma-rich, and Neuroendocrine-like (NE-like)[11].

While predictive UC biomarker characterization has mainly occurred in the context of checkpoint inhibitors (IMvigor 210/211[6,17], CheckMate 275[7], KEYNOTE-045[2]) and neoadjuvant chemotherapy[16], identification of predictive biomarkers for anti-angiogenic therapy has been lacking in clinical trials to date. Currently, the 2017 RANGE study is the only phase 3 trial to show a significant progression-free survival (PFS) benefit with the use of an angiogenesis inhibitor, ramucirumab, in combination with docetaxel after platinum-based therapy in the advanced/metastatic UC setting[18]. Follow-up analyses confirmed the PFS benefit, without a significant improvement in OS, in the intent-to-treat (ITT) population (OS 9.4 vs. 7.9 months, ramucirumab + docetaxel vs. docetaxel only)[19]. Intriguingly, exploratory biomarker analyses identified a strong association between higher baseline PD-L1 expression (combined positive score [CPS] ≥ 10) and increased OS with ramucirumab + docetaxel (hazard ratio [HR] 0.519, $p = 0.0048$, in patients with a PD-L1 CPS ≥ 10 vs. HR 0.999, $p = 0.9955$, in patients with a PD-L1 CPS < 10)[19]. These results suggest a putative role for predictive biomarkers of response to ramucirumab. Here, we report results from a retrospective analysis of the RANGE study using a comprehensive biomarker approach of PD-L1 status, tumor microenvironment gene expression signatures, and molecular subtyping, to identify patients who may optimally benefit from ramucirumab therapy for UC.

## Results

The ITT analysis population of the RANGE trial included 530 randomized (1:1) patients. Of these, 462 patient tumor samples were available to submit to Decipher Biosciences for gene expression profiling. Of these samples, 394 met assay criteria for gene expression profiling and 227 samples additionally met PD-L1 immunohistochemistry (IHC) assay criteria. Therefore, two cohorts comprised the translational research (TR) populations:

TR1 (gene expression profiling and PD-L1 IHC, $n = 227$) and TR2 (gene expression profiling, $n = 394$). Baseline demographics and disease characteristics of TR1/TR2 populations were representative of the overall RANGE ITT population with the exception of a lower proportion of East Asian patients in the TR1 (PD-L1) population (3.1% in the TR1 population vs. 20.8% in RANGE ITT and 18.8% in TR2) (Table 1). Both TR populations had significantly improved PFS for ramucirumab + docetaxel vs. placebo + docetaxel but not for OS, consistent with previous reports[18,19] for the RANGE ITT population (Table 1, Efficacy outcomes).

**PD-L1, tumor microenvironment gene expression signatures, and overall survival.** Archival patient tumor tissue from the RANGE trial was analyzed for PD-L1 IHC (22C3), according to the PD-L1 IHC 22C3 pharmDx Interpretation Manual for UC[20] (see Methods), to determine PD-L1 expression present in tumor cells (TC), immune cells (IC), and the CPS, which accounts for expression in both cell types. The CPS ≥ 10 cutoff was selected in line with the threshold defined in the PD-L1 IHC 22C3 pharmDx Interpretation Manual for UC[20] (see Methods). Higher PD-L1 CPS was associated with longer OS in the ramucirumab arm (CPS ≥ 10 vs. <10, median OS 9.03 vs. 7.92 months, Fig. 1a) of the TR1 population. In addition, CPS ≥ 10 was predictive of OS benefit for ramucirumab vs. placebo (stratified HR, 0.451, 95% confidence interval [CI] 0.275 to 0.74; $p = 0.002$, Benjamini-Hochberg [BH]-adjusted $p = 0.047$; Fig. 1a and Supplementary Table 1) compared to CPS < 10 (stratified HR, 0.926, 95% CI 0.604 to 1.42; interaction $p$-value = 0.063). Examination of the CPS ≥ 10 cutoff was also supported graphically with a sub-population treatment effect pattern plot (STEPP), by the observation that the 95% CI of the HR consistently appears below 1 when the subpopulation median CPS was approximately above 10 (Supplementary Fig. 1a). Tumor samples scored by PD-L1 expression in TC or IC similarly revealed an association between high-PD-L1 expression and longer OS in the ramucirumab arm (TC ≥ 1 vs. <1, 9.2 vs. 7.89 months, Fig. 1b; IC ≥ 4 vs. <4, 9.2 vs. 7.92 months, Fig. 1c). TC ≥ 1 and IC ≥ 4 trended towards OS benefit in the ramucirumab arm (Fig. 1b, c), although the interactions with treatment arm were not significant.

We subsequently examined angiogenesis and immune/inflammation pathways in the TR2 ($n = 394$) population. mRNA expression profiling revealed higher scores for published angiogenesis and immune signatures (see Methods) in tumor samples with PD-L1 CPS ≥ 10 vs. CPS < 10 (Fig. 2a). High PD-L1 on IC and TC both had considerable overlap with CPS ≥ 10 grouping and increased expression of angiogenesis and immune pathways. A significant proportion of the samples where PD-L1 status was not known were from the East Asia region and tended to have lower expression of both angiogenesis and immune signature.

Multiple angiogenesis and immune signature sets were summarized as a mean signature score (see Methods) to explore associations with both CPS and clinical outcomes. Higher angiogenesis and immune mean signature scores were associated with CPS ≥ 10 compared to CPS < 10 ($p < 0.01$ and $p < 0.0001$, respectively) (Fig. 2b). To analyze clinical outcomes associated with individual angiogenesis/immune signatures from the heat map (Fig. 2a), individual signature scores and mean signature scores were dichotomized by median, and subgroups were created by patients with scores >median or ≤median for each signature. Stratified HRs of ramucirumab + docetaxel vs. placebo + docetaxel with 95% CI were estimated using Cox regression models for each subgroup (Fig. 2c, Supplementary Tables 1 and 2). High immune signature scores provided OS ramucirumab benefit, with T-effector, T-cell

**Table 1 Demographics, baseline disease characteristics, and efficacy outcomes in the overall study ITT population compared to TR1 and TR2.**

| Variable, n (%) | ITT Population[a] N = 530 | | TR1 Population[b] N = 227 | | TR2 Population[c] N = 394 | |
|---|---|---|---|---|---|---|
| | Ramucirumab n = 263 | Placebo n = 267 | Ramucirumab n = 122 | Placebo n = 105 | Ramucirumab n = 198 | Placebo n = 196 |
| **Demographics and baseline disease characteristics** | | | | | | |
| **Gender** | | | | | | |
| Male | 213 (81.0) | 215 (80.5) | 105 (86.1) | 88 (83.8) | 166 (83.8) | 156 (79.6) |
| Female | 50 (19.0) | 52 (19.5) | 17 (13.9) | 17 (16.2) | 32 (16.2) | 40 (20.4) |
| Age, median years (range) | 65.0 (34–86) | 66.0 (32–83) | 64.5 (34–85) | 66.0 (32–79) | 65.0 (34–86) | 66.0 (32–83) |
| **Race group** | | | | | | |
| White | 203 (77.2) | 204 (76.4) | 116 (95.1) | 101 (96.2) | 159 (80.3) | 155 (79.1) |
| Asian | 54 (20.5) | 61 (22.8) | 4 (3.3) | 4 (3.8) | 36 (18.2) | 40 (20.4) |
| Other | 6 (2.3) | 2 (0.7) | 2 (1.6) | 0 (0.0) | 3 (1.5) | 1 (0.5) |
| **Region[d]** | | | | | | |
| North America | 24 (9.1) | 24 (9.0) | 10 (8.2) | 6 (5.7) | 20 (10.1) | 19 (9.7) |
| Europe/Other | 186 (70.7) | 186 (69.7) | 109 (89.3) | 95 (90.5) | 143 (72.2) | 138 (70.4) |
| East Asia | 53 (20.2) | 57 (21.3) | 3 (2.5) | 4 (3.8) | 35 (17.7) | 39 (19.9) |
| ECOG PS: %, 0 | 1[d, e] | 46.0 | 52.9 | 46.8 | 53.2 | 45.1 | 54.9 | 43.8 | 56.2 | 44.9 | 55.1 | 46.4 | 53.6 |
| **Primary tumor site** | | | | | | |
| Bladder | 180 (68.4) | 177 (66.3) | 92 (75.4) | 69 (65.7) | 135 (68.2) | 127 (64.8) |
| Non-bladder[f] | 72 (27.4) | 79 (29.6) | 25 (20.5) | 31 (29.5) | 56 (28.3) | 61 (31.1) |
| Other[g] | 11 (4.2) | 11 (4.1) | 5 (4.1) | 5 (4.8) | 7 (3.5) | 8 (4.1) |
| **Histology** | | | | | | |
| Pure transitional cell | 205 (77.9) | 217 (81.3) | 93 (76.2) | 81 (77.1) | 156 (78.8) | 156 (79.6) |
| Mixed | 55 (20.9) | 49 (18.4) | 29 (23.8) | 24 (22.9) | 42 (21.2) | 39 (19.9) |
| Missing | 3 (1.1) | 1 (0.4) | 0 (0.0) | 0 (0.0) | 0 (0.0) | 1 (0.5) |
| Baseline hemoglobin: %, <10 g/dL | ≥10 g/dL[h] | 12.9 | 85.2 | 13.5 | 85.8 | 10.7 | 89.3 | 17.1 | 82.9 | 10.6 | 89.4 | 15.8 | 83.2 |
| **Bellmunt risk factors[i]** | | | | | | |
| 0 | 88 (33.5) | 93 (34.8) | 40 (32.8) | 31 (29.5) | 65 (32.8) | 63 (32.1) |
| 1 | 105 (39.9) | 109 (40.8) | 51 (41.8) | 42 (40.0) | 81 (40.9) | 80 (40.8) |
| 2 | 64 (24.3) | 57 (21.3) | 26 (21.3) | 30 (28.6) | 46 (23.2) | 47 (24.0) |
| 3 | 6 (2.3) | 8 (3.0) | 5 (4.1) | 2 (1.9) | 6 (3.0) | 6 (3.1) |
| **Visceral metastases[d]** | | | | | | |
| Yes | 182 (69.2) | 188 (70.4) | 87 (71.3) | 83 (79.0) | 137 (69.2) | 147 (75.0) |
| No | 78 (29.7) | 79 (29.6) | 35 (28.7) | 22 (21.0) | 61 (30.8) | 49 (25.0) |
| Missing | 3 (1.1) | 0 (0) | 0 (0.0) | 0 (0.0) | 0 (0) | 0 (0) |
| Liver metastases present | 78 (29.7) | 69 (25.8) | 38 (31.1) | 31 (29.5) | 61 (30.8) | 56 (28.6) |
| **Prior platinum therapy** | | | | | | |
| Cisplatin | 161 (61.2) | 189 (70.8) | 73 (59.8) | 70 (66.7) | 125 (63.1) | 137 (69.9) |
| Carboplatin | 97 (36.9) | 77 (28.8) | 49 (40.2) | 34 (32.4) | 73 (36.9) | 58 (29.6) |
| Other | 2 (0.8) | 0 (0) | 0 (0) | 0 (0) | 0 (0) | 0 (0) |
| Missing | 3 (1.1) | 1 (0.4) | 0 (0) | 1 (1.0) | 0 (0) | 1 (0.5) |
| **Prior adjuvant therapy** | | | | | | |
| Adjuvant | 46 (17.5) | 70 (26.2) | 21 (17.2) | 24 (22.9) | 37 (18.7) | 54 (27.6) |
| Neo-adjuvant | 41 (15.6) | 37 (13.9) | 21 (17.2) | 14 (13.3) | 31 (15.8) | 31 (15.8) |
| None | 173 (65.8) | 160 (59.9) | 80 (65.6) | 67 (63.8) | 130 (65.7) | 111 (56.6) |
| Missing | 3 (1.1) | 0 (0) | 0 (0) | 0 (0) | 0 (0) | 0 (0) |
| **Efficacy outcomes** | | | | | | |
| Median PFS, months (95% CI) | 4.07 (3.29–4.83) | 2.76 (2.60–2.89) | 4.37 (4.01–5.32) | 2.69 (1.54–2.83) | 4.17 (3.65–5.22) | 2.73 (1.97–2.89) |
| HR (stratified), (95% CI) | 0.70 (0.58–0.86) | | 0.60 (0.44–0.81) | | 0.64 (0.51–0.81) | |
| p-value (log-rank) Stratified | <0.001 | | <0.001 | | <0.001 | |
| Median OS, months (95% CI) | 9.40 (7.89–11.43) | 7.85 (7.00–9.30) | 8.48 (6.77–11.14) | 6.11 (4.50–7.43) | 8.80 (7.59–11.43) | 7.06 (5.85–7.92) |
| HR (stratified), (95% CI) | 0.88 (0.72–1.08) | | 0.75 (0.55–1.02) | | 0.83 (0.66–1.05) | |
| p-value (log-rank) Stratified | 0.212 | | 0.068 | | 0.130 | |

CI confidence interval, ECOG PS Eastern Cooperative Oncology Group Performance Status, HR hazard ratio (of treatment effect), IHC immunohistochemistry, ITT intent-to-treat population, N total number of patients in corresponding arm and population, n number of patients in specified category, OS overall survival, PD-L1 programmed cell death ligand 1, PFS progression-free survival, TR translational research.
[a] ITT population consists of all randomized patients from the original RANGE study; baseline characteristics have been published previously[18,19].
[b] TR1 population consists of patients in the ITT population from whom both PD-L1 IHC and valid Decipher Biosciences RNA results were obtained.
[c] TR2 population consists of patients for whom valid Decipher Biosciences RNAseq results were obtained.
[d] Stratification factors.
[e] ECOG PS data missing for 3 patients in the ramucirumab arm of the ITT population.
[f] Non-bladder primary tumor site refers to renal pelvis, ureter, or urethra.
[g] "Other" refers to tumors with more than one primary site.
[h] Baseline hemoglobin values missing for 5 and 2 patients in the ramucirumab and placebo arm respectively, of the ITT population.
[i] Bellmunt risk factors included liver metastases, hemoglobin <10 g/dL, and ECOG PS score > 0.

inflamed, activated CD4, activated CD8, memory CD8, and mean immune signatures showing OS treatment effect (HRs ranging from 0.574 to 0.699 [Supplementary Table 2], BH-adjusted $p < 0.2$; Supplementary Table 1. Interactions between treatment and immune signature scores (>median vs. ≤median) were significant for the T-effector ($p = 0.036$) and activated CD4 T-cell ($p = 0.022$) (Fig. 2c). However, there was no clear association between high angiogenesis signature scores and ramucirumab benefit (Fig. 2c). Additionally, to explore the association between ramucirumab benefit and angiogenesis and immune mean signature score in a continuous manner, the STEPP method was utilized (Supplementary Fig. 1b, c). The 95% CI of the HR was consistently below 1 when the subpopulation median immune signature score went above 0.36 confirming the finding in dichotomized analysis that high immune signature scores provided OS ramucirumab benefit. However, no clear trend was observed for the angiogenesis mean signature score.

**Urothelial carcinoma molecular subtype, PD-L1 status, and overall survival.** We analyzed the mRNA dataset using three different gene expression-based UC molecular classifiers[11,21] (Supplementary Table 3). There was considerable overlap between ConsensusMIBC Basal/Squamous and the Basal and Claudin Low subtypes from Decipher Bioscience's Genomic Subtyping Classifier Bladder version 1 (GSCv1). The ConsensusMIBC classifier further refines the Basal subtype from Decipher GSCv1 into an even split between Basal/Squamous and Stroma-rich subtypes. Analysis of the association between molecular subtype and PD-L1 CPS revealed that Basal tumor types of classification schemes were more likely to be CPS ≥ 10: 52.2% of Basal and 82.1% of Claudin Low subtypes using the Decipher GSCv1 classifier[21], 73.5% of the Basal/Squamous subtype using the ConsensusMIBC classifier[11] (Fig. 3a), and 76.9% of the Basal Squamous subtype using TCGA were CPS ≥ 10. In contrast, Luminal tumor types of UC classification schemes were more

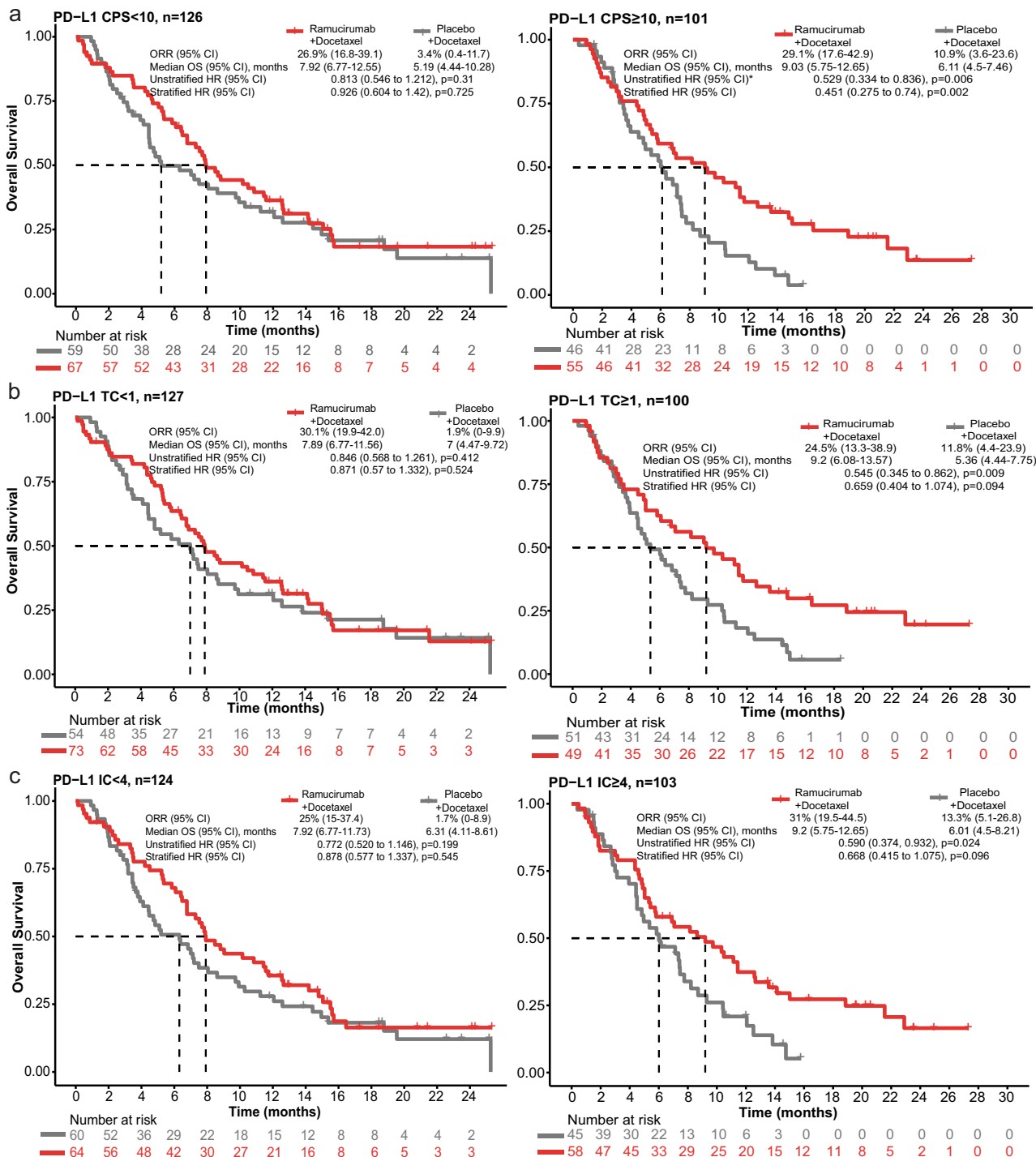

**Fig. 1 Biomarker association with clinical outcome: PD-L1 and signature pathways. a–c** Kaplan–Meier curves depicting OS probability in ramucirumab + docetaxel or placebo + docetaxel arms based on PD-L1 expression. PD-L1 scoring method and cutoffs scored by **a** CPS < 10 ($n = 126$ participants) vs. ≥10 ($n = 101$ participants), **b** TC < 1 ($n = 127$ participants) vs. ≥1 ($n = 100$ participants), and **c** IC < 4 ($n = 124$ participants) vs. ≥4 ($n = 103$ participants). ORR, median OS, and stratified/unstratified HRs are shown. Stratification was based on geographical region, baseline ECOG PS, and visceral metastases. A two-sided Wald test was used in Cox regression models. *p*-values before BH-adjustment are shown in figures. *p*-values after BH-adjustment are shown in Supplementary Table 1. For all models, the TR1 population ($n = 227$ participants) is used, and the number within each subset is reported above. *Indicates proportional hazard assumption was violated with $p = 0.04$. Source data are provided as a Source Data file. BH, Benjamini-Hochberg; CI, confidence interval; CPS, combined positive score; ECOG PS, Eastern Cooperative Oncology Group Performance Status; HR, hazard ratio; IC, immune cell; n, number of participants; ORR, objective response rate; OS, overall survival; PD-L1, programmed cell death ligand 1; TC, tumor cell; TR, translational research.

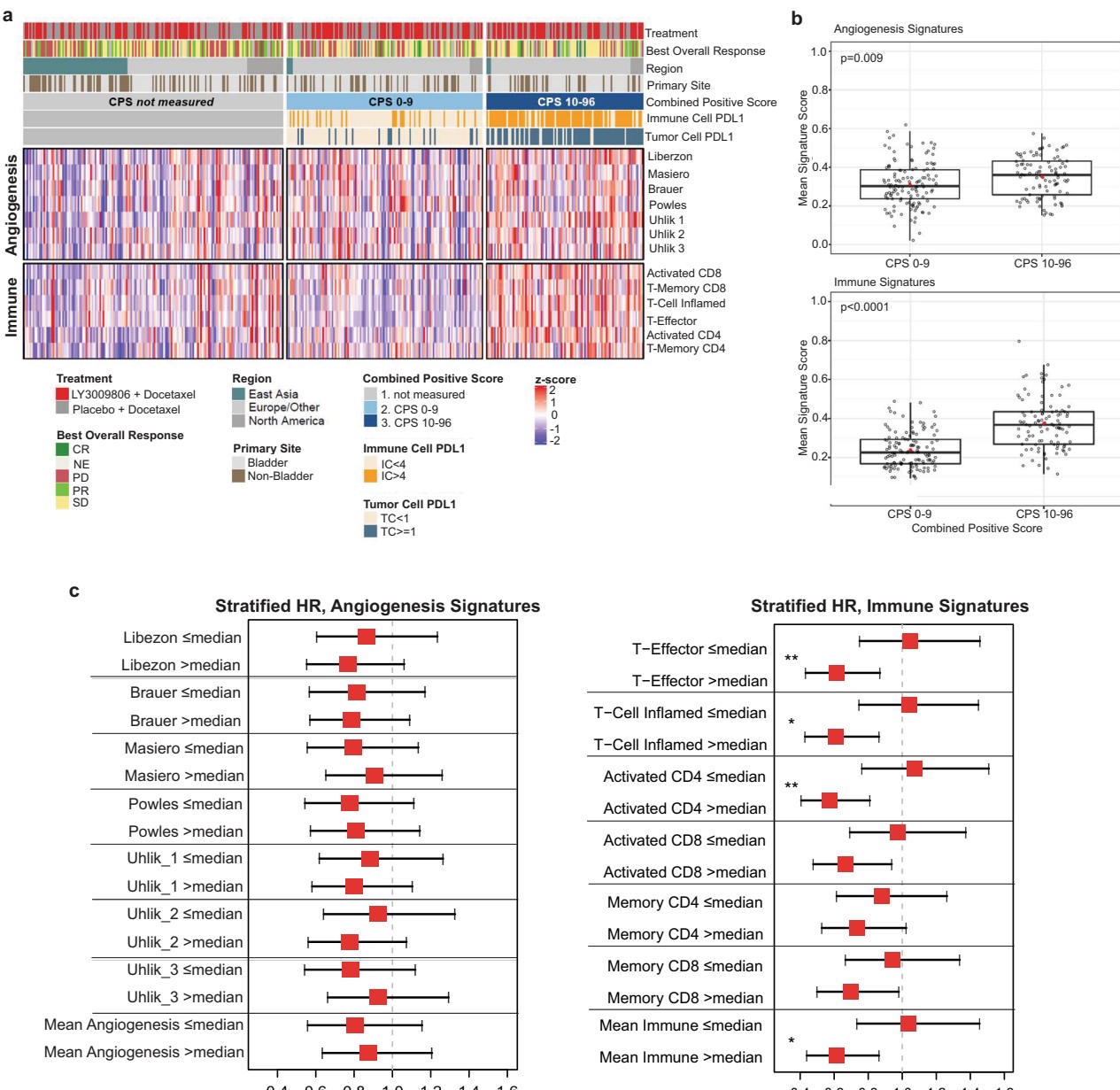

**Fig. 2 Molecular subtype association with PD-L1 status, angiogenesis/immune signatures, and clinical outcome. a** Heatmap of angiogenesis and immune gene signatures in the TR2 population (*n* = 394 participants). Columns (patient samples) are ordered by CPS PD-L1 group (CPS not measured, CPS 0–9 and CPS 10–96) and by region (East Asia, Europe/Other, and North America). **b** Mean signature score for angiogenesis and immune pathways in CPS < 10 (*n* = 126) vs. CPS ≥ 10 (*n* = 101) samples from *n* = 227 participants in the TR1 population. For boxplots, the center line represents median, box hinges represent first and third quartiles, whiskers represent minimum and maximum within 1.5x interquartile range, and red marker is mean. Mean angiogenesis signature score, *p* = 0.009 (CPS < 10 vs. CPS ≥ 10, two-sample *t*-test without multiplicity adjustment). Mean immune signature score, *p* < 0.0001 (CPS < 10 vs. CPS ≥ 10, two-sample *t*-test without multiplicity adjustment). **c** Forest plot of stratified OS HRs (95% CIs) of ramucirumab + docetaxel vs. placebo + docetaxel for the subgroups defined by dichotomized individual angiogenesis and immune signatures, and corresponding mean of angiogenesis and immune signature score. Data are presented as estimated HR with error bars indicating the 95% CI. Each signature is dichotomized by the median. Citations for the angiogenesis signature pathways can be found in the Methods and Supplementary Table 2. Stratification was based on geographical region, baseline ECOG PS, and visceral metastases. The proportional hazard assumption was not violated in any instance. The TR2 population (*n* = 394 participants) was used; *n* = 197 participants within each subgroup. *Interaction *p*-value < 0.1 (0.055 for T-cell inflamed; 0.063 for mean immune); **Interaction *p*-value < 0.05 (0.036 for T-effector; 0.022 for activated CD4). *p*-values were based on two-sided Wald test without multiplicity adjustment. Source data are provided as a Source Data file. CI, confidence interval; CPS, combined positive score; CR, complete response; HR, hazard ratio; IC, immune cell; n, number of participants; NE, not evaluable; PD, progressive disease; PD-L1, programmed cell death ligand 1; PR, partial response; SD, stable disease; TC, tumor cell.

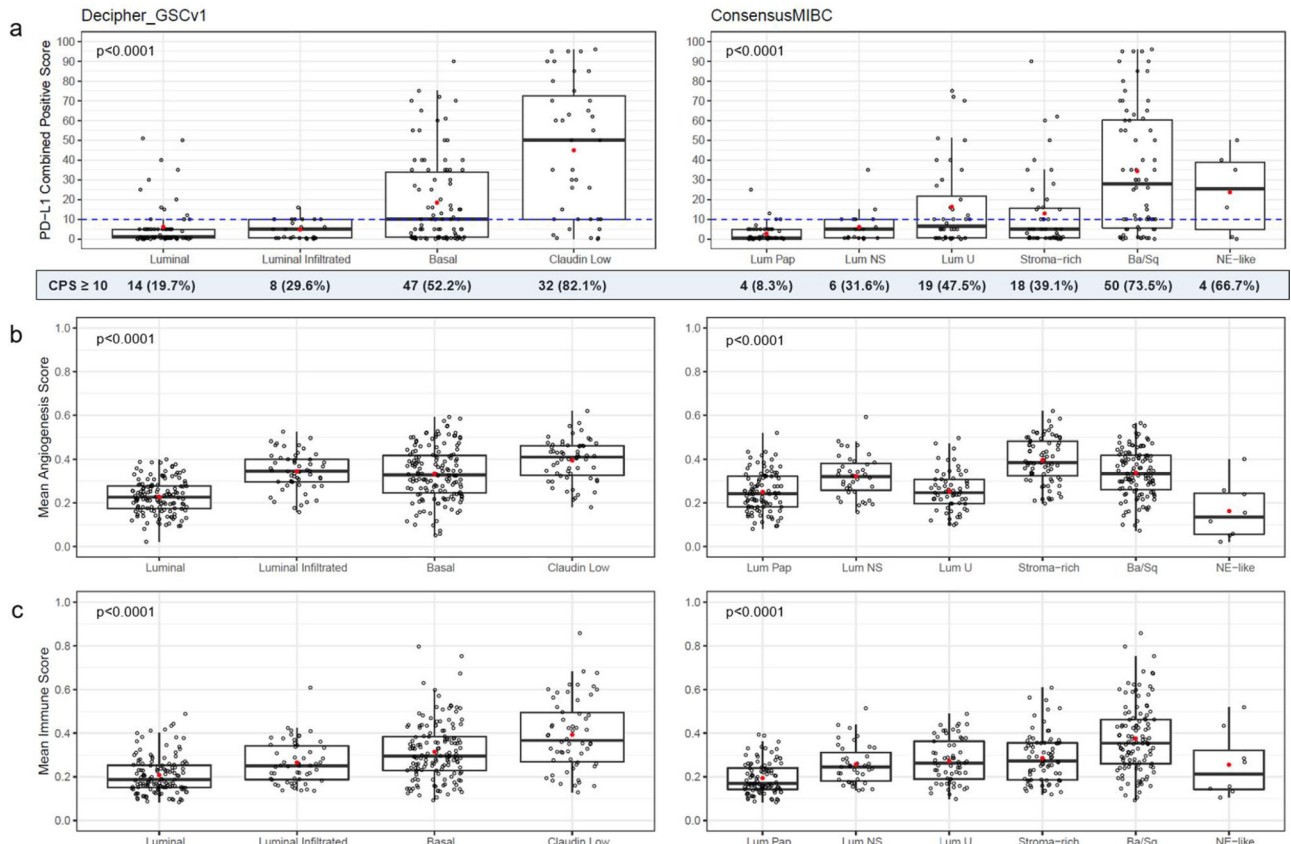

**Fig. 3 Molecular subtype association with PD-L1 CPS and mean angiogenesis/immune signature scores. a** Association of PD-L1 expression by CPS with molecular subtypes as defined by the Decipher GSCv1 and ConsensusMIBC classification schemes in the TR1 population ($n = 227$ samples from $n = 227$ participants in the TR1 population). For boxplots, the center line represents median, box hinges represent first and third quartiles, whiskers represent minimum and maximum within 1.5x interquartile range, and red marker is mean. Decipher GSCv1 F-test, $p < 0.0001$; ConsensusMIBC F-test, $p < 0.0001$. $p$-values indicated are for one-way ANOVA, without multiplicity adjustment. The blue dotted line indicates CPS ≥ 10 cutoff. Number and percent of patients with CPS ≥ 10 for each tumor subtype indicated below the plot. **b** Mean of angiogenesis signature scores relative to molecular subtypes of the Decipher GSCv1 and ConsensusMIBC classification schemes in the TR2 population ($n = 394$ samples from $n = 394$ participants in the TR2 population). Decipher GSCv1 F-test, $p < 0.0001$; ConsensusMIBC F-test, $p < 0.0001$, without multiplicity adjustment. $p$-values indicated are for one-way ANOVA. **c** Mean of immune signature scores relative to molecular subtypes of the Decipher GSCv1 and ConsensusMIBC classification schemes in the TR2 population ($n = 394$ samples from $n = 394$ participants in the TR2 population). Decipher GSCv1 F-test, $p < 0.0001$; ConsensusMIBC F-test, $p < 0.0001$. $p$-values indicated are for one-way ANOVA, without multiplicity adjustment. For boxplots, the center line represents median, box hinges represent first and third quartiles, whiskers represent minimum and maximum within 1.5x interquartile range, and red marker is mean. Decipher GSCv1 subtype prevalence ($n$/group): Luminal ($n = 131$), Luminal Infiltrated ($n = 55$), Basal ($n = 150$), Claudin Low ($n = 58$). ConsensusMIBC subtype prevalence ($n$/group): Luminal Papillary ($n = 97$), Luminal Non-Specified ($n = 39$), Luminal Unstable ($n = 61$), Stroma-rich ($n = 75$), Basal/Squamous ($n = 114$), Neuroendocrine-like ($n = 8$). Source data are provided as a Source Data file. Ba/Sq, Basal/Squamous; CPS, combined positive score; Lum NS, Luminal Non-Specified; Lum Pap, Luminal Papillary; Lum U, Luminal Unstable; NE-like, Neuroendocrine-like; PD-L1, programmed cell death ligand 1.

often CPS < 10. For example, 80.3% of Luminal and 70.4% of Luminal Infiltrated using the Decipher GSCv1 classifier and 91.7% (Luminal Papillary), 68.4% (Luminal Non-Specified), and 52.5% (Luminal Unstable) subtypes using the ConsensusMIBC classifier were CPS < 10 (Fig. 3a). A chi-square test for CPS ≥ 10 vs. Decipher GSCv1 subtypes was significant ($p < 0.0001$). A Fisher's exact test for CPS ≥ 10 vs. ConsensusMIBC subtypes was also significant ($p = 0.0005$). In addition, Basal and Claudin Low of Decipher GSCv1 subtypes and Basal/Squamous of ConsensusMIBC classification were associated with higher CPS levels when it was investigated as a continuous variable (Fig. 3a). With respect to expression signatures, the Claudin Low and Basal Decipher GSCv1 molecular subtypes had high mean angiogenesis and immune signature scores, while the Luminal subtype had the lowest scores (Fig. 3b, c). The Basal/Squamous subtype within the ConsensusMIBC classification had the second highest mean angiogenesis signature score and highest mean immune signature

score (Fig. 3b, c). The Stroma-rich subtype had the highest angiogenesis score, while Luminal Papillary had the lowest score for both mean angiogenesis (excluding $n = 8$ NE-like) and immune signature score.

Analysis of OS outcomes based on Decipher GSCv1 molecular subtype revealed that patients with Basal type tumors had longer median OS with ramucirumab treatment over placebo (Basal, 10.48 vs. 7.75 months; Claudin Low, 8.48 vs. 5.65 months), with the Claudin Low subtype exhibiting the strongest ramucirumab treatment effect (stratified HR 0.508, 95% CI 0.259 to 0.994, $p = 0.048$, BH-adjusted $p = 0.226$; Fig. 4 and Supplementary Table 1). Of note, the Claudin Low subtype also had the highest mean angiogenesis and immune signature scores (Fig. 3b, c). Luminal subtypes, which had the lowest overall mean angiogenesis/immune signature scores, showed the least treatment benefit. With the ConsensusMIBC classification scheme, the Basal/Squamous subtype benefited most from ramucirumab (stratified

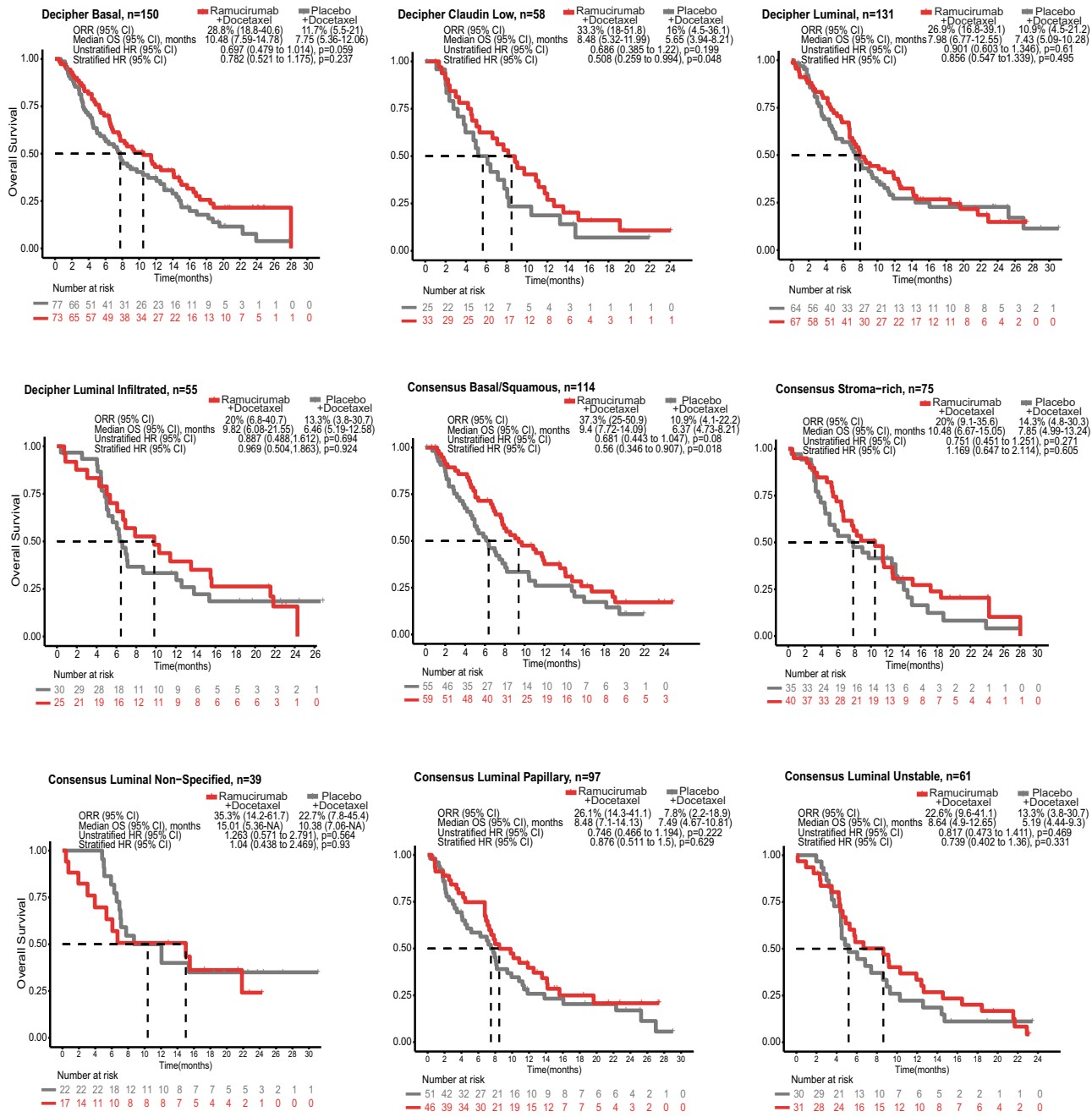

**Fig. 4 Molecular subtype association with clinical outcome.** Kaplan–Meier curves representing OS probability in ramucirumab + docetaxel or placebo + docetaxel arms based on Decipher GSCv1 molecular subtypes (Basal [*n* = 150 participants], Claudin Low [*n* = 58 participants], Luminal [*n* = 131 participants], and Luminal Infiltrated [*n* = 55 participants]) or ConsensusMIBC (Basal/Squamous [*n* = 114 participants], Stroma-rich [*n* = 75 participants], Luminal Non-Specified [*n* = 39 participants], Luminal Papillary [*n* = 97 participants], and Luminal Unstable [*n* = 61 participants]) molecular subtype. Neuroendocrine-like subtype was not analyzed due to low number of patient samples for this subtype (*n* = 8). ORR, median OS, and stratified/unstratified HRs are shown. A two-sided Wald test was used in Cox regression. *p*-values before BH-adjustment are shown in figures. *p*-values after BH-adjustment are shown in Supplementary Table 1. Stratification was based on geographical region, baseline ECOG PS, and visceral metastases. TR2 population (*n* = 394 participants) is used. The proportional hazard assumption was not violated in any instance. Source data are provided as a Source Data file. BH, Benjamini-Hochberg; CI, confidence interval; ECOG PS, Eastern Cooperative Oncology Group Performance Status; HR, hazard ratio; n, number of participants; ORR, objective response rate; OS, overall survival; TR, translational research.

HR 0.56, 95% CI 0.346 to 0.907, *p* = 0.018, BH-adjusted *p* = 0.134) (Fig. 4 and Supplementary Table 1). The association of OS with Basal/Squamous subtype was also consistent when using a TCGA 2017 classification (Supplementary Fig. 2). The Luminal Papillary subtype, which had the lowest overall mean signature score for both angiogenesis (excluding *n* = 8 NE-like)

and immune signatures across Basal and Luminal subtypes (Fig. 3b, c), showed reduced treatment benefit compared to the Basal/Squamous subtype. The interaction between treatment and Decipher GSCv1 subtypes (Claudin Low vs. others) was not significant (*p* = 0.32), while the interaction between treatment and ConsensusMIBC subtypes (Basal/Squamous vs. others) was

marginally insignificant ($p = 0.058$). Interestingly, while the Stroma-rich subtype had the highest angiogenesis score, with relatively lower immune scores, there was no differential ramucirumab OS benefit. Supplementary Fig. 3a, b show the angiogenesis/immune signature heatmap grouped by molecular subtype and CPS. Figure 5 summarizes ramucirumab treatment benefit for OS based on inferred cell type from both classification schemes, confirming the greatest ramucirumab treatment benefit is seen in Basal type tumors and least benefit in tumors with Luminal characteristics. The Stroma-rich subtype from the ConsensusMIBC classifier showed the least benefit overall. The ConsensusMIBC Basal/Squamous grouping showed greater benefit compared to Basal type tumors as defined by Decipher GSCv1 (Fig. 5).

**Multivariable analysis of clinical factors associated with expression signatures, molecular subtypes, and ramucirumab benefit**. We examined multivariable analyses investigating the association of OS outcomes with mean angiogenesis (> vs. ≤median) subtypes, mean immune (> vs. ≤median) subtypes, (Decipher GSCv1 and ConsensusMIBC separately), and their interactions with treatment, using stratified Cox regression with stepwise selection. The interaction between mean immune signature and ramucirumab was selected and significant in the model selecting variables with Decipher GSCv1 (Supplementary Table 4a, $p = 0.047$). The interaction between ConsensusMIBC and ramucirumab was selected and significant in the model selecting variables with ConsensusMIBC (Supplementary

Table 4b, $p = 0.035$). We further explored the association between mean angiogenesis and immune signature scores and clinical covariates. Multivariable analysis showed significantly lower scores for both angiogenesis and immune signature score for patients from the East Asia geographic region vs. other regions after adjusting for multiple clinical covariates (Fig. 6a, b; Supplementary Table 5). We assessed whether this could be attributed to a higher fraction of luminal molecular subtypes in the patients enrolled from East Asia. There were slightly higher percentages (by ~9%) of participants with Luminal (Decipher GSCv1) or Luminal Papillary (ConsensusMIBC) subtypes in the East Asia geographic region compared to all other regions. This small difference suggests that this discrepancy was not entirely attributable to differences in proportion of molecular subtypes (Fig. 6c). We further compared mean angiogenesis and immune signature score across molecular subtypes, now in relation to geographic region, and observed that the East Asia subgroup had numerically lower mean immune and angiogenesis signature scores across all bladder cancer molecular subtypes, as defined by either Decipher GSCv1 or ConsensusMIBC (Fig. 7a, b). Together, this suggests the existence of different tumor microenvironments in patient subgroups within this trial, defined by geographic location. The presence of lower mean angiogenesis and immune signature score were also associated with reduced benefit of ramucirumab + docetaxel, as patients in the East Asian geographic subgroup did not show as much OS benefit with ramucirumab + docetaxel (Fig. 7c and Supplementary Table 1) compared to other regions. This indicates that disease heterogeneity related to geographic region may have contributed to the OS difference in this study. Finally, the presence of visceral metastases (defined in this study as liver, lung, or bone) correlated with a significantly higher mean angiogenesis signature score (Fig. 6a; Supplementary Fig. 4).

## Discussion

The RANGE trial randomized platinum-refractory UC patients to receive ramucirumab + docetaxel or placebo + docetaxel without biomarker or molecular subtype selection. While positive for PFS, RANGE failed to show a significant OS impact for the overall patient population. In the current study, we investigated PD-L1 expression, analyzed immune and angiogenesis pathways relevant to the mechanism of action of ramucirumab, and correlated UC molecular subtypes with OS outcomes in UC. We found that high-PD-L1 expression in either tumor cells or tumor-associated immune cells, upregulation of immune signatures, and Basal molecular subtypes were associated with ramucirumab OS benefit. It is important to note that, as described in the Results section of this manuscript, the baseline disease and demographic characteristics of the two translational research populations (TR1 and TR2) were largely representative of the overall RANGE ITT population.

Disparate methods used for assigning PD-L1 status, including the use of different antibodies and staining platforms for IHC, different PD-L1 scoring metrics (CPS[2] vs. TC[9] vs. IC[17]), and non-standardized cutoffs for defining high vs. low, have complicated interpretation of the role PD-L1 plays as a potential biomarker across UC studies[22]. Here, we took the approach to examine PD-L1 expression present in TC, IC, and CPS. Our results (Fig. 1a, b, c) suggest high-PD-L1 expression (in both the immune and tumor cell compartments) is associated with greater ramucirumab benefit in UC. Therefore, measuring PD-L1 in both cell types may be clinically justified, as expression in both IC and TC are potential determinants of ramucirumab + docetaxel benefit.

The anti-angiogenic mechanism of ramucirumab as an antitumor agent led us to examine gene expression pathways related to

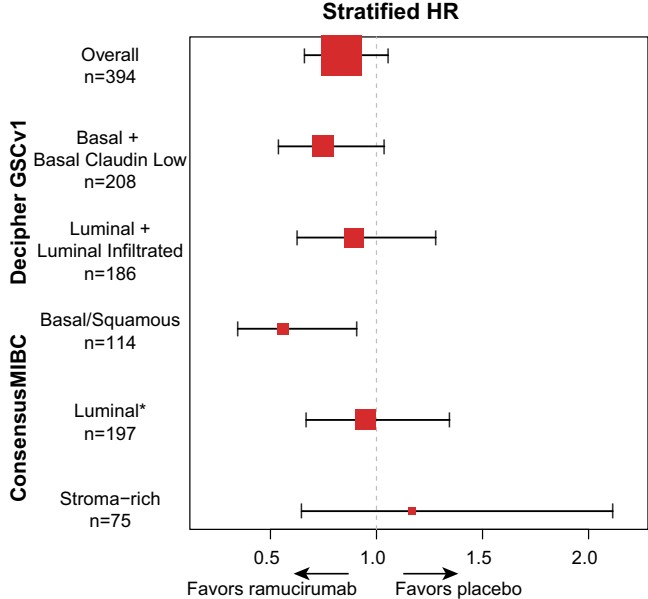

**Fig. 5 Forest plot of stratified HRs (95% CIs) for overall survival based on inferred cell type from both molecular subtype classifications.** The ConsensusMIBC Neuroendocrine subtype had too few observations ($n = 8$ participants) for a stand-alone analysis and was not included. Data are presented as estimated HR with error bars indicating the 95% CI and box size indicating the sample size. The TR2 population ($n = 394$ participants) is used. Number of participants for each subgroup are: Basal + Basal Claudin Low, $n = 208$ participants; Luminal + Luminal Infiltrated, $n = 186$ participants; Basal/Squamous, $n = 114$ participants; Luminal*, $n = 197$ participants; and Stroma-rich, $n = 75$ participants. *"Luminal" grouping within the ConsensusMIBC classification includes Luminal Papillary, Luminal Unstable, and Luminal Non-Specified. Source data are provided as a Source Data file. CI, confidence interval; HR, hazard ratio; n, number of participants.

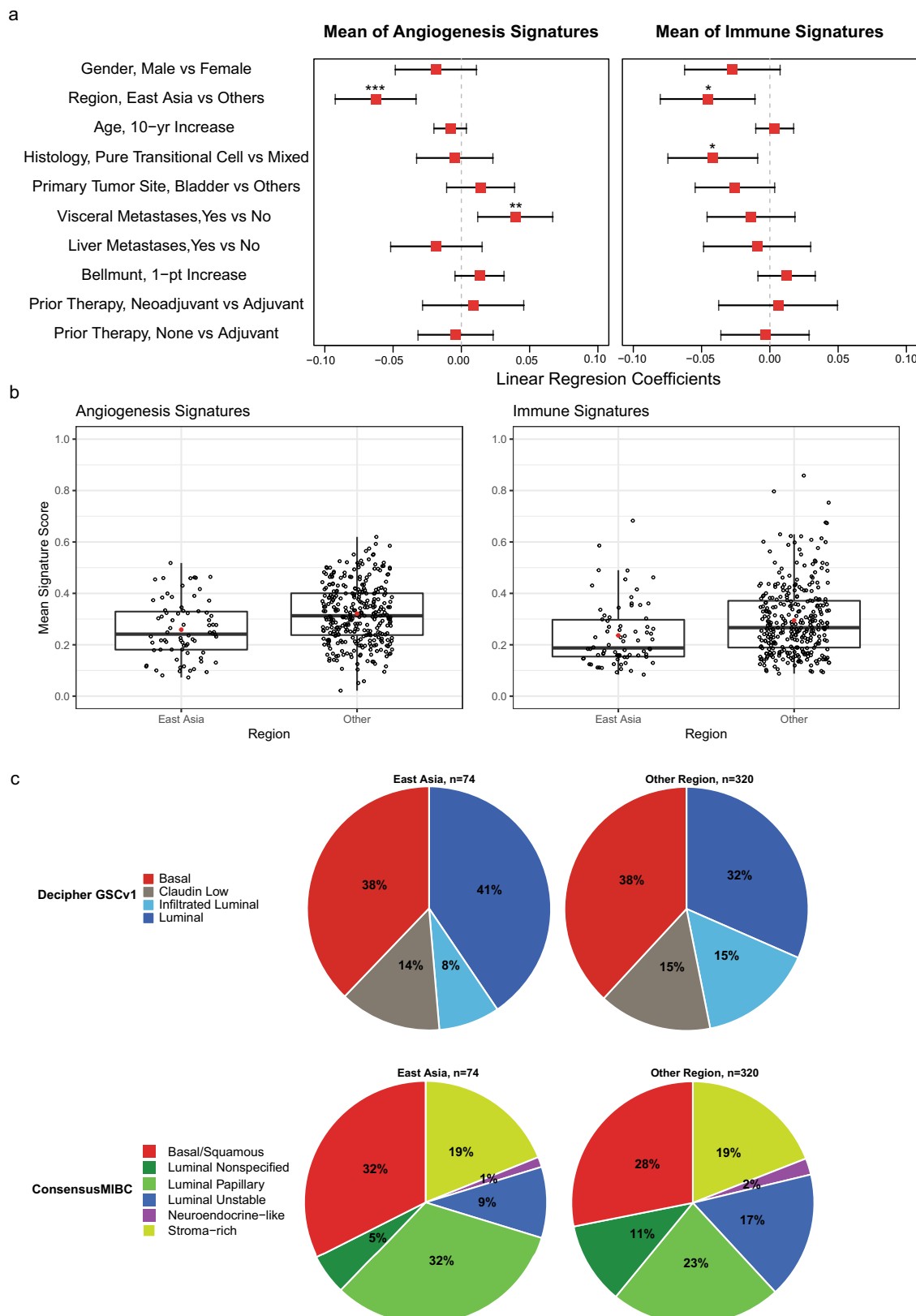

angiogenesis as potential markers for response. In addition, the exploratory analysis from the RANGE trial[19], showing that PD-L1 CPS ≥ 10 was associated with significant OS treatment effect, prompted us to further investigate immune-related gene signatures. Our analysis revealed that high-PD-L1 expression correlated with upregulation of both angiogenesis and immune-related signatures (Fig. 2a, b), but only high immune signature score was associated with significant ramucirumab OS benefit (Fig. 2c).

We used two single-sample prediction methods for UC molecular subtyping, Decipher GSCv1 and ConsensusMIBC, with

**Fig. 6 Identification of clinical covariates associated with angiogenesis- and immune- mean signature score and relation to clinical outcome. a** The coefficients estimated by a multivariable linear regression with indicated clinical covariates for the mean of angiogenesis or mean of immune signature score, respectively. Data are presented as mean estimated coefficient with error bars indicating the 95% CI. For categorical covariates with two levels (i.e., gender, region, histology, primary tumor site, visceral or liver metastases, and prior therapy), the coefficient represents the expectation of the mean signature score of the first category subtracted by the expectation of the second category as listed on the y-axis. For Bellmunt risk factor and age, the coefficient represents the slope of the mean signature score per 1-point increase in number of risk factors or 10-years age, respectively, as noted on the y-axis. Data are shown for n = 394 participants from the TR2 population. Significant associations are denoted by *, **, *** corresponding to *p = 0.01, **p = 0.005, ***p < 0.001 (t-test in linear regression without multiplicity adjustment; exact p-values are shown in Supplementary Table 5). **b** Mean angiogenesis and immune signature score in relation to geographic region (East Asia n = 74 vs. Other n = 320) in the TR2 population (n = 394 participants). Mean of angiogenesis signature score, p < 0.0001 (East Asia vs. Other, two-sample t-test, equal variance); mean of immune signature score, p < 0.001 (East Asia vs. Other, Welch two-sample t-test, unequal variance). For boxplots, the center line represents median, box hinges represent first and third quartiles, whiskers represent minimum and maximum within 1.5x interquartile range, and red marker is mean. **c** Proportion of molecular subtypes from the Decipher GSCv1 and ConsensusMIBC classification schemes across geographic region (East Asia n = 74 vs. Other n = 320). Pie charts show percentage of participants for indicated molecular subtype in the TR2 population (n = 394 participants). Source data are provided as a Source Data file. pt, point; yr, year.

similarities and differences noted across the reported molecular subtypes, PD-L1 score, and immune and angiogenesis signatures. This allowed us to provide a more comprehensive assessment of the association of these factors with survival outcomes. We noted that high PD-L1 (CPS ≥ 10) was differentially associated with molecular subtypes, with a relatively higher number of cases in Basal subtypes across both subtyping schemes. When comparing outcomes with the Decipher GSCv1 classifier, patients with Basal and Claudin Low subtypes (i.e., those with relatively higher mean immune/angiogenesis signature scores and higher PD-L1 expression) had relatively larger improvement in median OS with ramucirumab + docetaxel vs. placebo + docetaxel compared to those with a luminal subtype, which displayed the lowest PD-L1 and mean immune/angiogenesis signature scores. With the ConsensusMIBC classification, it is notable that Basal/Squamous and Stroma-rich subtypes (corresponding to Decipher GSCv1 Basal and Claudin Low) showed upregulated mean angiogenesis signature scores, but only Basal/Squamous had a particularly high mean immune signature score relative to the remaining subtypes. This correlated with a relatively larger improvement in median OS for the Basal/Squamous subgroup compared to the Stroma-rich subtype with ramucirumab + docetaxel, suggesting that presence of immune cells in the tumor microenvironment may be needed. Intriguingly, in renal cell carcinoma, an in-depth analysis of the IMmotion150 study showed that the combination of bevacizumab and atezolizumab was particularly effective in the T-effector high population compared to the atezolizumab monotherapy or sunitinib arms[23]. This association was confirmed in the phase 3 IMmotion151 trial, showing a better outcome for the atezolizumab/bevacizumab arm compared to sunitinib in the T-effector/proliferative subtype[24]. Thus, the molecular subgroups found to respond well to ramucirumab in our study could be particularly amenable to combinations of an anti-angiogenic and immunotherapeutic treatment. Further work exploring anti-angiogenic agents and their interactions with the immune and stromal microenvironment in UC is needed. Overall, our results suggest that PD-L1 expression, immune and angiogenesis signatures, and molecular subtypes may predict for response to ramucirumab.

In the last few years, there have been increasing efforts and evidence building towards the use of molecular subtyping in UC for potential use in clinical management; for example, prognostic association of subtypes and predicting response to neoadjuvant chemotherapy[16] and genomic correlates of response to PD-1/PD-L1 inhibitors (IMvigor210[21], CheckMate 275[25], and PURE-01[21]). Some of this work has been hampered by a lack of standardization of approach; the use of both Decipher and ConsensusMIBC single-sample classifiers are a step towards filling that gap. The

results of the current study provide some unique insights; for example, the Basal subtype (ConsensusMIBC), a subtype with poor prognosis[11], showed improved outcomes with ramucirumab + docetaxel. Although not yet formally tested, Luminal type tumors, which are enriched for fibroblast growth factor receptor 3 (FGFR3) genomic alterations[25], may respond better to FGFR pathway inhibitors, such as erdafitinib[26]. Our multivariable analysis of clinical covariates associated with angiogenesis and immune pathways identified patients from the East Asia region as having significantly lower scores for both pathways, which also correlated with a lower ramucirumab treatment effect. Of note, East Asian participants had a slightly higher proportion of luminal type subgroups (Fig. 6c): 41% of patients (30 of 74) from East Asia, vs. 32% of patients (101 of 320) from other regions. Furthermore, East Asian participants had a significantly higher proportion of non-bladder primary site of disease: 54.1% of patients (40 of 74) from East Asia vs. 28.8% of patients (92 of 320) from other regions.

Many anti-angiogenic approaches have been tested in UC based upon promising initial data only to fail to meet endpoints in later development[27]. RANGE was a phase 3 trial of an anti-angiogenic agent in UC, achieving positive results for the primary endpoint of PFS; however, a statistically significant improvement in OS was not shown. This retrospective analysis provides possible explanations for the failure to demonstrate OS benefit in an unselected population and suggests potential biomarkers to inform patient selection in future UC anti-angiogenic therapy trial designs. Almost half of the patients in this study had Luminal type tumors. If extrapolated back to the RANGE ITT population, this may have potentially diluted the therapeutic effect since we have now shown data suggesting that patients most likely to benefit from ramucirumab treatment are those with Basal tumor types. Future studies could explore our findings in additional UC trial datasets using anti-angiogenic therapy or anti-angiogenic/PD-1 directed therapy combinations, such as CALGB 90601 (addition of bevacizumab to frontline platinum-based therapy)[28] and the currently enrolling LEAP-011 trial[29] (pembrolizumab with or without lenvatinib in platinum-ineligible PD-L1 CPS-high patients). Importantly, in CALGB 90601, PFS benefit was reported, but no OS benefit was reported when anti-angiogenic therapy was added to chemotherapy in a biomarker unselected patient population[28]. Biomarker work in these trials may confirm our findings and potentially lead to novel treatment strategies involving anti-angiogenic and immune-modulating therapies in selected populations.

Our study has several limitations. This was a retrospective study and most tumor samples used in this analysis were archival and not from biopsies taken immediately prior to enrollment in

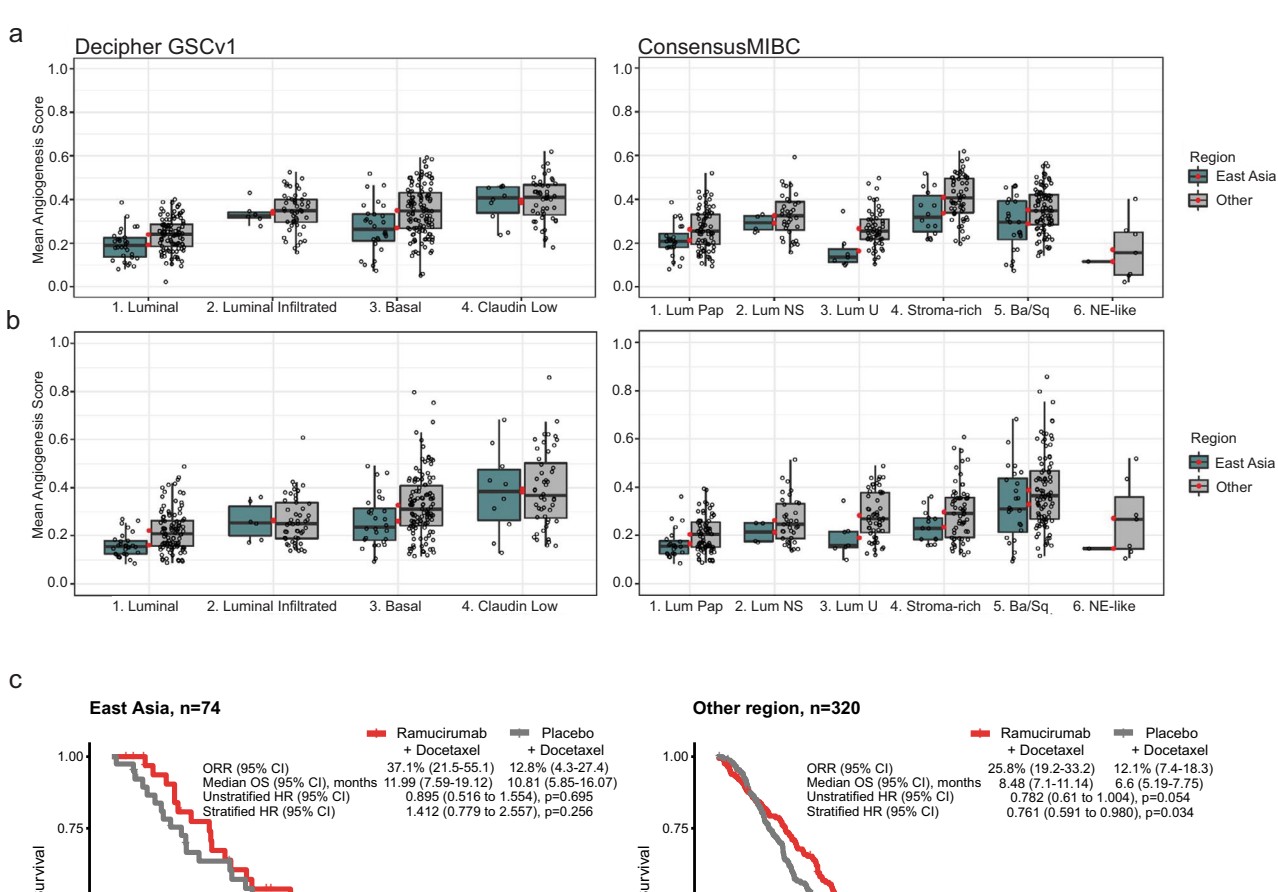

**Fig. 7 Molecular subtype association with mean angiogenesis/immune signature scores and geographical region. a, b** Mean of angiogenesis (**a**) and immune (**b**) signature scores, by geographic region, relative to molecular subtypes of the Decipher GSCv1 and ConsensusMIBC classification schemes in the TR2 population ($n = 394$ samples from $n = 394$ participants in the TR2 population). Decipher GSCv1 subtype prevalence (n/group; East Asia/Other): Luminal ($n = 131$; $n = 30/n = 101$), Luminal Infiltrated ($n = 55$; $n = 6/n = 49$), Basal ($n = 150$; $n = 28/n = 122$), Claudin Low ($n = 58$; $n = 10/n = 48$). ConsensusMIBC subtype prevalence (n/group; East Asia/Other): Luminal Papillary ($n = 97$; $n = 24/n = 73$), Luminal Non-Specified ($n = 39$, $n = 4/n = 35$), Luminal Unstable ($n = 61$; $n = 7/n = 54$), Stroma-rich ($n = 75$; $n = 14/n = 61$), Basal/Squamous ($n = 114$; $n = 24/n = 90$), Neuroendocrine-like ($n = 8$; $n = 1/n = 7$). For boxplots, center line represents median, box hinges represent first and third quartiles, whiskers represent minimum and maximum within 1.5x interquartile range, and red marker is mean. **c** Kaplan–Meier curves representing OS probability in ramucirumab + docetaxel or placebo + docetaxel arms based on geographic region. ORR, median OS, and stratified/unstratified HRs are shown. Two-sided Wald test was used in Cox regression. p-values before BH-adjustment are shown in figures. p-values after BH-adjustment are shown in Supplementary Table 1. Stratification was based on geographical region, baseline ECOG PS, and visceral metastases. TR2 population ($n = 394$ participants) is shown; there were $n = 74$ participants included in the East Asia subgroup and $n = 320$ participants included in the other regions subgroup. The proportional hazard assumption was not violated in any instance. Source data are provided as a Source Data file. BH, Benjamini-Hochberg; Ba/Sq, Basal/Squamous; CI, confidence interval; ECOG PS, Eastern Cooperative Oncology Group Performance Status; HR, hazard ratio; Lum NS, Luminal Non-Specified; Lum Pap, Luminal Papillary; Lum U, Luminal Unstable; n, number of participants; NE-like, Neuroendocrine-like; ORR, objective response rate; OS, overall survival; TR, translational research.

RANGE. Thus, there could have been changes in molecular subtype and/or PD-L1 status related to prior treatment. Additionally, heterogeneity of the tumor tissue sampled by the biopsy and further by the macrodissection process may impact the molecular subtype classification. Given the assay requirements of PD-L1 and the age of the tumor samples (in some cases only slides were available), only a subset of the RANGE tumors could

be assessed for PD-L1, leading to a smaller patient population in TR1, as well as fewer East Asian samples with PD-L1 data. Furthermore, because East Asian sites enrolled early, a disproportionate number of samples from that region were past the six-month window for evaluation of PD-L1. As such, the interaction of molecular subtypes and PD-L1 status in East Asian patients with UC remains to be completely characterized. Decipher

GSCv1 and ConsensusMIBC subtype classifiers have not formally been assessed in the context of differences related to geography or non-bladder primary site tumor locations. Further work is needed to understand this potential association.

While the results reported herein are hypothesis generating and limited by sample size, they represent an important step in linking OS response to anti-angiogenic therapy with biomarkers in UC. Owing to the retrospective and exploratory nature of the analyses presented, appropriate caution is warranted when interpreting strength of associations. However, given the overlapping nature of the Basal molecular subtypes and PD-L1-high status to predict sensitivity to ramucirumab, further exploration of these signatures is warranted to identify patient subsets most likely to benefit from anti-angiogenic therapy approaches, as well as anti-angiogenic and PD-1/PD-L1 combination approaches in UC and other tumor types.

## Methods

**Ethics approval**. The trial complied with the Declaration of Helsinki, the International Conference on Harmonization Guidelines for Good Clinical Practice, and applicable local regulations. The protocol was approved by the ethics committees of all participating centers (for a full list of investigators and participating centers, see Supplementary Table 6), and all patients provided written informed consent.

**Study design**. The RANGE trial (clinicaltrials.gov: NCT02426125) was a randomized, double-blind, phase 3 study evaluating ramucirumab in combination with docetaxel in patients with platinum-refractory advanced or metastatic UC. The overall study design, protocol, and procedures have been reported[18]. Key eligibility criteria included patients with cytologically or histologically confirmed transitional cell UC originating from the bladder, urethra, ureter, or renal pelvis; locally advanced, unresectable, or metastatic disease extent; and documented disease progression 14 months or less after platinum-containing chemotherapy in the adjuvant or first-line metastatic setting. Prior treatment with one immune-checkpoint inhibitor was permitted. For these patients, ≤24 months from the end of the platinum-containing regimen was allowed to accommodate the additional immune-checkpoint inhibitor line of therapy. Patients were ineligible if they had received more than one systemic chemotherapy regimen in the relapsed or metastatic setting.

**Study procedures**. Eligible patients were randomized (1:1) to receive intravenous ramucirumab 10 mg/kg or placebo, both in combination with intravenous docetaxel 75 mg/m$^2$ (60 mg/m$^2$ in Korea, Taiwan, or Japan) on day 1 of a 21-day cycle (i.e., every 3 weeks). Randomization was stratified by geographic region (North America, East Asia, Europe/rest of the world), Eastern Cooperative Oncology Group Performance Status (ECOG PS) at baseline (0 or 1), and visceral metastasis (yes or no), with visceral metastases being liver, lung, and/or bone. Docetaxel was limited to 6 cycles, with up to 4 additional cycles permitted upon approval by sponsor; ramucirumab (or placebo) treatment continued until at least one discontinuation criterion was met. Dose modifications (reductions or delays) were permitted according to protocol-defined criteria.

The post hoc, exploratory analysis of biomarker associations with clinical outcomes examined OS, defined as the time from date of randomization to the date of death from any cause. If the patient was alive at the data cutoff date for analysis (or was lost to follow-up), OS data were censored for analysis on the last date the patient was known to be alive.

**Sample collection and immunohistochemistry**. In the RANGE study, if pretreatment tumor tissue specimens were available, submission of archived tissue was mandatory unless restricted by local regulations. For assessing PD-L1 status, sample slides that were within 6 months from sectioning were processed using a commercially available automated IHC assay (PD-L1 IHC 22C3 pharmDx, Agilent/Dako; catalog number SK006)[20], on the Dako Autostainer Link 48 platform. The PD-L1 IHC 22C3 pharmDx primary antibody and visualization reagent are prediluted in ready-to-use format. If more than 6 months had passed since sectioning, those sample slides were considered out of the stability window and were excluded from analyses for purposes of this exploratory biomarker study. Scoring was performed by a blinded independent central lab (NeoGenomics Laboratories, Inc) using the following three methods: (1) CPS score, as described in the PD-L1 IHC 22C3 pharmDx Interpretation Manual for UC[20], assesses PD-L1 expression in both tumor and immune cells. CPS is calculated by the number of PD-L1-stained cells (tumor cells, lymphocytes, and macrophages) divided by the total number of viable tumor cells then multiplied by 100. Specimens were categorized as PD-L1 high if CPS ≥ 10 and PD-L1 low if CPS < 10, in line with the threshold described in the PD-L1 IHC 22C3 pharmDx Interpretation Manual for UC and the registration

for pembrolizumab in first-line cisplatin ineligible locally advanced or metastatic UC[20]. The CPS ≥ 10 cutoff was additionally assessed graphically using the STEPP method[30]. (2) Similar to the tumor proportion score, tumor cell (TC) score, described in the PD-L1 IHC 22C3 pharmDx Interpretation Manual for non-small cell lung cancer[31], is the percentage of viable tumor cells showing partial or complete PD-L1 membrane staining at any intensity relative to all viable tumor cells present in the sample. Samples were categorized by median into TC ≥ 1 and TC < 1. 3) Immune cell (IC) score, which is the percentage of tumor area occupied by PD-L1-expressing immune cells[32]. Samples were dichotomized at the median IC value into PD-L1 high and low. This was IC ≥ 4 for high and PD-L1 low if IC < 4, closely reflecting a 5% IC cutoff used in the Roche Ventana PD-L1 (SP142) Assay.

**Gene expression data generation**. Tumor samples from 462 patients in the RANGE phase 3 trial were submitted to Decipher Biosciences (previously GenomeDx) for tumor sample gene expression profiling and UC molecular subtyping. A sufficient number of slides ($N = 10$) per tumor sample with 5 μM thick tissue sections were provided to Decipher for expression profiling with $n = 1$ slide used for H&E staining to assess tumor cellularity. H&E guided macrodissection was done to achieve at least 0.5 mm$^2$ with a tumor percent macrodissected area of ≥50% to control for tumor purity. Transcriptome analysis was performed on formalin-fixed, paraffin-embedded tumor tissue with GeneChip® Human Exon 1.0 ST Array (Affymetrix) in a Clinical Laboratory Improvement Amendments-certified laboratory. Expression data were generated on 410/462 samples with 32 samples failing for low tumor content, one failing due to low RNA quantity, and 19 failing at cDNA preparation stage. Of 410 samples, 394 passed microarray quality control (16 failed) and were included in these analyses. Microarray data were normalized using single-channel array normalization (SCAN)[33] are accessible through GEO Series accession number GSE198269.

**Pathway signature and analysis**. We used established relevant angiogenesis and immune signature collections to better understand their association with the expression profiles in the $n = 394$ population. These signatures were: (a) MSigDB angiogenesis hallmark[34]; (b) VEGF-dependent vasculature genes associated with response to anti-VEGF therapy[35]; (c) primary tumor angiogenesis signature[36]; (d) endothelial-specific stromal phenotype signatures[37]; (e) molecular signatures of microvascular endothelium; (f) core angiogenesis pathway genes[23]; and (g) signatures related to immune-infiltration, effector function[38–40] and activated/memory CD4/CD8 T cells[41].

Gene signature scores were calculated as the mean of the log2 SCAN normalized gene expression values for genes comprising the signatures, and corresponding z-scores were generated for heatmaps. Signatures were selected based on postulated related biology, and the seven angiogenesis and six immune signatures showed significant, positive intercorrelations within each set ($p < 0.001$). In addition, the average of each signature set showed significant, positive correlations with the individual signatures comprising the set (angiogenesis $r^2$ 0.76–0.95, $p < 0.001$; immune $r^2$ 0.65–0.95, $p < 0.001$; Supplementary Fig. 5). The signature set average was included for analysis as single angiogenesis and immune score and evaluated relative to results obtained with individual signatures.

**Urothelial carcinoma molecular subtype**. Decipher Bioscience's GSCv1 is a consensus of pre-existing The Cancer Genome Atlas (TCGA)[10,15,42] (2014 and 2017) and University of North Carolina (UNC)[14,43] classifiers. It uses 149 markers to assign tumor expression profiles to four main classes (Basal, Claudin Low, Luminal Infiltrated, and Luminal)[16,21]. Decipher provided GSCv1 classifier derived bladder cancer subtype calls and TCGA 2017 subtype calls[42] on 394 samples for the RANGE trial.

Additionally, a more recent consensus molecular classifier derived by the Bladder Cancer Molecular Taxonomy Group was used[11]. ConsensusMIBC (muscle-invasive bladder cancer) is based on consensus of six classification schemes (Baylor[44], UNC[43], MD Anderson Cancer Center[13], Lund[45], Cartes d'Identité des Tumeurs-Curie[46], and TCGA[15]) and assigns tumor expression profiles to six consensus classes: Basal/Squamous, Luminal Papillary, Luminal Non-Specified, Luminal Unstable, Stroma-rich, and NE-like. Subtype calls on 394 samples were made using the ConsensusMIBC package (version 1.0) provided by the authors (https://github.com/cit-bioinfo/consensusMIBC).

**Complex heatmap**. Heatmaps were generated using the ComplexHeatmap package (version 1.99.8)[47]. Heatmap annotations were organized into separate blocks of related parameters including: study variables (treatment arm, best overall response, region/geographical location, and primary tumor site); PD-L1 IHC scoring variables (PD-L1 CPS, IC, and TC); and UC molecular subtyping classification schemes including TCGA 2014[10], ConsensusMIBC classifier[11], and Decipher GSCv1[21]. SCAN normalized angiogenesis and immune signature z-scores were plotted for the 394 expression cohort samples. Heatmap columns (i.e., patient samples) were ordered by PD-L1 IHC CPS score groups (CPS not measured, CPS 0–9, CPS 10–96), region, and/or by Decipher GSCv1 or ConsensusMIBC molecular subtype.

**Statistical analysis**. In this post hoc, exploratory analysis, biomarker cohorts were defined as follows: translational research 1 (TR1) cohort consisted of 227 patient samples that had both expression profiling and valid PD-L1 IHC data, and TR2 cohort consisted of 394 patients for whom biomarker expression data were available. Demographics and baseline disease characteristics were summarized for the ITT ($n = 530$) population and both TR populations ($n = 227$ and $n = 394$). Of note, both TR populations are a subset of the ITT population, and TR1 is a subset of TR2.

Overall survival by treatment arms was estimated with Kaplan–Meier curves for each biomarker-defined subgroup. Subgroups were defined by patients with PD-L1 CPS ≥ 10 and CPS < 10; PD-L1 TC ≥ 1 and TC < 1; PD-L1 IC ≥ 4 and IC < 4; immune/angiogenesis signature scores >median and ≤median; and Decipher GSCv1 and ConsensusMIBC subtypes. For each of the specified subgroups, hazard ratios and p-values were calculated for the ramucirumab arm relative to the placebo arm using Cox regression. Proportional hazard assumptions were checked using the score test based on weighted residual[48] and reported for all Cox regression models. When reporting results of stratified Cox regression, clinical trial stratification factors, including geographical region, baseline ECOG PS, and visceral metastases, were used to adjust for possible imbalance across treatment arms within each biomarker subgroup or molecular subtype. Forest plots of stratified HR were used to summarize the treatment effect for subgroups defined by angiogenesis/immune scores or by subtypes. In addition, interaction p-values between treatment arm and biomarkers (including PD-L1 CPS ≥ 10 vs. CPS < 10, immune/angiogenesis signature scores >median vs. ≤median, and subtypes) were calculated using stratified Cox regression models.

Continuous measurements for PD-L1 CPS, angiogenesis, and immune mean signature scores were examined using the STEPP method[30] to determine if the magnitude of the treatment effect changed as a function of the values of these biomarkers. STEPP analyses help determine whether the treatment effect changes for subpopulations with different biomarker values. This is done by dividing the total patient population into overlapping subpopulations based on different biomarker thresholds (known as the sliding window approach), estimating the treatment effect within each subpopulation, and plotting treatment effects against biomarker values. In PD-L1 CPS analysis, 60 patients were included in each window, and there were 40 patients in common between two consecutive windows. In analyses of angiogenesis and immune mean signatures, 120 patients were included in each window, and there were 60 patients in common between two consecutive windows. HRs were plotted against median biomarker values in each window.

Objective response rate was defined as the proportion of randomized patients within each subgroup achieving a best overall response of complete response or partial response per Response Evaluation Criteria in Solid Tumors (RECIST) version 1.1[49]. The association between clinical covariates and mean signature score was assessed using a multivariable linear regression model for both angiogenesis and immune signature, respectively. The coefficients of linear regression models were estimated for all clinical covariates according to the following regression model and reported:

$$Y_i = \beta_0 + \beta_1 I\{Gender_i = Male\} + \beta_2 I\{Region_i = East\ Asia\} + \beta_3 Age_i + \beta_4 I\{Histology_i = Pure\ Transitional\ Cell\} + \beta_5 I\{Primary\ Tumor\ Site_i = Bladder\} + \beta_6 I\{Visceral\ Metastases_i = Yes\} + \beta_7 I\{Liver\ Metastases_i = Yes\} + \beta_8 Bellmunt_i + \beta_9 I\{Prior\ Therapy_i = Neoadjuvant\} + \beta_{10} I\{Prior\ Therapy_i = None\} + \varepsilon_i$$

where $Y_i$ represents angiogenesis mean signature score in the left panel and immune mean signature score in the right panel of patient $i$ (Fig. 6a); $\beta_0, \ldots, \beta_{10}$ are the coefficients; I{} represents the indicator function for categorical variables; and $\varepsilon_i$ denotes the random error of patient $i$.

For hypotheses with a single comparison, a two-sided 5% significance level has been used. For the hypothesis of superior treatment effect from ramucirumab, multiple comparisons were conducted in specified subgroups. BH-adjusted[50] p-values were calculated and reported to account for multiple comparisons. Owing to the post hoc, exploratory nature of the analyses, BH-adjusted p-values below 20% have been highlighted throughout the manuscript for interpretation purposes. Statistical analyses of clinical data were performed using SAS 9.4 or R 4.1.0.

## Data availability

The RANGE gene expression data generated in this study have been deposited in NCBI's Gene Expression Omnibus[51] and are accessible through GEO Series accession number GSE198269. The RANGE clinical-trial data generated in this study have been deposited at www.vivli.org. RANGE clinical-trial data are available under restricted access to protect patient privacy, and access can be obtained by submitting a request. For details on submitting a request, see the instructions provided at www.vivli.org. For specific study details, see here: https://search.vivli.org/?search=NCT02426125. Through www.vivli.org, Eli Lilly and Company will provide access to all de-identified, individual-level participant data collected during the trial except for pharmacokinetic or genetic data. Data are available to request 6 months after the indication studied has been approved in the USA and EU or after primary publication acceptance, whichever is later. No expiration date of data requests is currently set once data have been made available. Access to data will be provided after a proposal has been approved by an independent review committee

identified for this purpose and after receipt of a signed data sharing agreement. On average, it takes 2–3 months to review a request to access data on the Vivli platform. This is the time from request submission to approval by the Independent Review Committee. Multiple factors can impact the timeline to access the data, including the number of data contributors, the number of studies, the availability of the requestor to respond to comments, the ability to align with the data use agreement, and if the data from the trial have already been anonymized. Data and documents, including the study protocol, statistical analysis plan, clinical study report, and blank or annotated case report forms, will be provided in a secure data sharing environment. Source data for figures are provided with this paper. Source data are provided with this paper.

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

## Acknowledgements

This study was funded by Eli Lilly and Company. The funder provided support in the form of salaries for authors (RRH, ERR, MSX, RR, SW, KMB-M, and AA), design of study, data collection, data analysis, decision to publish, and preparation of the manuscript. We thank the patients, their families, and the study personnel across all sites for participating in the RANGE trial. We also thank Decipher Biosciences for performing the gene expression profiling and molecular subtyping of the tumor samples, as well as insightful discussions (Elai Davicioni and Ewan Gibb). Kristi Gruver and Susan Whitman of Eli Lilly and Company provided project management support. Erika Wittchen and Antonia Baldo of Syneos Health provided medical writing and editorial support, funded by Eli Lilly and Company.

## Author contributions

Conception and/or design: M.S.v.d.H., T.P., D.P., R.d.W., C.N.S., R.R.H., K.M.B.-M., A.A., A.D. Data analysis and/or interpretation: M.S.v.d.H., T.P., D.P., R.d.W., A.N., C.N.S., N.M., H.N., D.C., S.A.H., G.G., J.L.L., S.T.T., U.V., J.B.A.-C., R.R.H., E.R.R., M.S.X., R.R., S.W., K.M.B.-M., A.A., A.D. Data acquisition: M.S.v.d.H., R.d.W., A.N., C.N.S., N.M., H.N., A.B., G.G., J.L.L., S.T.T., U.V., J.B.A.-C., B.J.E., R.R.H., E.R.R., A.A., A.D. Drafting and/or critical revision of the manuscript: M.S.v.d.H., T.P., D.P., R.d.W., A.N., C.N.S., N.M., H.N., D.C., S.A.H., A.B., G.G., J.L.L., S.T.T., U.V., J.B.A.-C., B.J.E., R.R.H., E.R.R., M.S.X., R.R., S.W., K.M.B.-M., A.A., A.D. All authors read and approved the final manuscript: M.S.v.d.H., T.P., D.P., R.d.W., A.N., C.N.S., N.M., H.N., D.C., S.A.H., A.B., G.G., J.L.L., S.T.T., U.V., J.B.A.-C., B.J.E., R.R.H., E.R.R., M.S.X., R.R., S.W., K.M.B.-M., A.A., A.D.

## Competing interests

M.S.v.d.H. has received research support from Bristol Myers Squibb, AstraZeneca, Roche, and 4SC; and consultancy fees from BMS, Merck, Sharp & Dhome, Roche, AstraZeneca, Seattle Genetics, Janssen, and Pfizer (all paid to Institute). T.P. reports consulting honoraria from AstraZeneca, BMS, Exelixis, Incyte, Ipsen, Merck, MSD, Novartis, Pfizer, Seattle Genetics, Merck Serono, Astellas, Johnson & Johnson, Eisai, and Roche; grants/funding (to Institution) from AstraZeneca, BMS, Exelixis, Ipsen, Merck, MSD, Novartis, Pfizer, Seattle Genetics, Merck Serono, Astellas, Johnson & Johnson, Eisai, and Roche; and travel expenses from Roche, Pfizer, MSD, AstraZeneca, and Ipsen. D.P. has received consultancy fees from Ada Cap (Advanced Accelerator Applications) Amgen, Astellas, AstraZeneca, Bayer, Bicycle Therapeutics, Boehringer Ingelheim, Bristol Myers Squibb, Clovis Oncology, Eli Lilly and Company, Exelixis, Gilead Sciences, Incyte, Janssen, Mirati, Monopteros, Pfizer, Pharmacyclics, Regeneron, Roche, Seattle Genetics, and Urogen; grant support from Ada Cap (Advanced Accelerator Applications), Agensys Inc., *Astellas, AstraZeneca, *Bayer, BioXcel Therapeutics, Bristol Myers Squibb, Clovis Oncology, Eisai, *Eli Lilly and Company, *Endocyte, Genentech, *Innocrin, Med-Immune, Medivation, Merck, Mirati, *Novartis, Pfizer, *Progenics, Replimune, Roche, *Sanofi Aventis, and Seattle Genetics (*denotes study trials have terminated); and ownership interest/investment in Bellicum (sold 7/2020) and Tyme (sold 10/2019). R.d.W. has received consultancy fees from Merck, Sanofi, Astellas, Janssen, Clovis, Orion, and Bayer; speaker fees from Astellas, Sanofi, and Merck; and has received research grants (to Institution) from Sanofi and Bayer. A.N. has a consulting role with Merck, AstraZeneca, Janssen, Incyte, Roche, Rainier Therapeutics, Clovis Oncology, Bayer, Astellas/Seattle Genetics, Ferring, and Immunomedics; has received grant/research support from Merck, Ipsen, and AstraZeneca; and travel expenses/honoraria from Roche, Merck, AstraZeneca, and Janssen. C.N.S. has served in an advisory/consultancy role for Pfizer, MSD, Merck, AstraZeneca, Astellas, Sanofi-Genzyme, Roche-Genentech, Incyte, Medscape, UroToday, and Foundation Medicine. N.M. reports personal fees from Chugai, Janssen, MSD, and Sanofi and grants from Astellas Pharma, Inc., AstraZeneca, Chugai, Eli Lilly and Company, Taiho, Janssen, MSD, Takeda, Amgen, and Pfizer. H.N. reports personal fees from Astellas Pharma, Inc., Chugai, Janssen, Merck, Sharp & Dhome, and Nippon Kayaku and grants from Astellas Pharma, Inc., Ono, and Chugai. S.A.H. has an advisory/consulting role with Roche, MSD, AstraZeneca, BMS, Janssen, Astellas, Bayer, Ipsen, Pfizer, Pierre Fabre, and Sotio and has received research funding from Cancer Research UK, MRC/NIHR, Janssen, and Boehringer Ingelheim. A.B. has an advisory/consulting role with Roche, Pfizer, Bristol Myers Squibb, AstraZeneca, and IPSEN Pharma; receiving honoraria from Bristol Myers Squibb, IPSEN Pharma, and Merck, Sharpe & Dohme and research support from AstraZeneca, Bristol Myers Squibb, Pfizer, IPSEN Steering Committee, and Roche. G.G. has served on an Advisory Board (last 5 years) for Bayer, MSD, Medac, IPSEN Pharma, Merck, LEO Pharma, and Astellas and has acted as an expert for Erbe Elektromedizin, IPSEN Pharma, Roche, Ferring, LEO Pharma, Merck, and Medac. S.T.T. reports research funding (to Institution) from Eli Lilly and Company and Sanofi and honoraria from Sanofi. U.V. reports research support from Astellas, Merck, and Bristol Myers Squibb and consulting and honoraria from AAA Pharmaceutical, Aveo, Bristol Myers Squibb, Bayer, Exelixis, Merck, Pfizer, and Sanofi. J.B.A.-C. serves on the Speakers' Bureau of Astellas/Seattle Genetics and Bristol Myers Squibb and receives advisory board fees from EMD Serono, Pfizer, Aveo Pharmaceuticals, Immunomedics, AstraZeneca, Exelixis, and Merck. B.J.E. serves in an advisory/consultancy role for Merck, Janssen, AstraZeneca, Roche, Pfizer, EMD Serono, and SEAGEN. R.R.H., E.R.R., M.S.X., and S.W. are employees and shareholders of Eli Lilly and Company. K.M.B.-M. is a shareholder and former employee of Eli Lilly and Company and a shareholder and current employee of AbbVie. A.A. is a shareholder and former employee of Eli Lilly and Company and is currently an employee of Daiichi Sankyo, Inc. (US). A.D. has an advisory/consulting role with AstraZeneca, PACT Pharma, Astellas/Seattle Genetics, Janssen, Nektar, Bristol Myers Squibb, Radmetrix,

Merck, Roche/Genentech, Exelixis, and Dyania Health; has received travel reimbursement from Eli Lilly and Company, AstraZeneca, and Seattle Genetics; holds stock and/or other interests in Attica Sciences and ATHOS Therapeutics; and has received research funding (to Institution unless noted otherwise) from AstraZeneca, Genentech/Roche, BMS, Merck Sharp & Dohme, Jounce Therapeutics, Infinity Pharmaceuticals, Seattle Genetics/Astellas, and Kite/Gilead (to A.D.). The disclosures reported herein, for each author, are potential conflicts only on the premise that the companies listed manufacture drugs for cancer treatment. The following authors declare no competing interests: D.C., A.B., J.L.L., R.R.
