## [Peer Review File · Nature Communications]

Reviewers' Comments:

Reviewer #1:

Remarks to the Author:

van der Heijden and colleagues have undertaken a retrospective correlative study of samples from the RANGE trial (which reported positive for PFS and negative for OS). Both transcriptome (Decipher) and PD-L1 IHC data is generated, and these are used to identify associations between subgroups and response, and between epidemiologic features and molecular ones. The statistical and bioinformatics analysis appears to be poor, with many lacking controls and incorrect analyses -- but the Methods are very terse, and it's possible many analyses are correct and just poorly reported. Changes in the Methods could significantly change the reporting, potentially leading to removal of statements around subtype-specificity and others, which would significantly change the impact of this report.

1. Statistics

- * Adjustments for multiple-testing are needed throughout, for example Figure 1a-c gives six separate p-values while figure 1f gives 15 and figure 3a gives 20
- * CPS discretization threshold was not rationalized
- * CPS is continuous (or at least integer) and should be analyzed as such throughout
- * Statistical methods are insufficiently detailed
- * the repeated $p=2.00e-16$ values likely reflect numerical estimation limits, not true p-values tied with exactly that value
- * the statistics and reporting of subtypes do not match. The statistics provide a one-way ANOVA (i.e. means are not all equal) while the text discusses "likelihood to be CPS < 10" and "subgroup with the second highest score". These concepts are not statistically equivalent: either restate the text to match the analysis, or update the analysis to match the hypothesis being considered.
- * similarly, the text discusses which subtypes "benefited most from ramucirumab", but this cannot be determined from the numerical comparison of HRs, this requires a statistical comparison of them (e.g. a likelihood ratio test or (better) an interaction term). Numerous statements suffer from this problem, for example "confirming the greatest ramucirumab treatment benefit is seen in basal type tumors". And of course multiple-testing needs to be appropriately adjusted for in these analyses.
- * outline the way in which the PH assumption was verified for all Cox models, and specifically state which (if any) models showed violations: it's surprising with this many models that none are reported
- * multivariate modeling rather than repeated univariate modeling is needed in Figure 3 to identify a region effect independent of other covariates

2. Bioinformatics

- * the module eigengene (ME) score is not interpretable in its current form. It appears that a PCA was done on ~7 angiogenesis signatures and separately on ~6 immune signatures, then the loadings of the first PC of each was used as a summary measure. There does not appear to be any QC of the PCA, nor an indication of percent variance explained, nor an analysis of the correlation structure of the scores, and so forth. There is no rationale given for why a PCA is superior to simply taking the median or mean score
- * Bioinformatics methods are insufficiently detailed
- * No clear controlling for tumour purity in transcriptome analyses, for example using deconvolution algorithms

3. Data Availability

- * The DECIPHER data comprises a key component of the value of this study, and needs to be made available both to readers and reviewers.

Reviewer #2:

Remarks to the Author:

This manuscript describes a biomarker study embedded in the RANGE study that evaluated Ramucirumab for treatment of patients with platinum refractory bladder cancer. Please consider

the following comments:

Introduction

1) Line 77-78 – the use of the term “intrinsic” is not uniformly agreed upon as it relates to these subtypes. Suggest deleting this term

2) Line 91-93 – I am not aware that a “numeric improvement” is meaningful when the OS endpoint is non-significant. Please rephrase noting that OS was non-significant without reference to “numerical improvement”

Results

3) Lines 103-105 – please include the total number of samples submitted to Decipher so we can understand the % that passed QC for both subtype and POD-L1. 227 ends up representing only 43% of the randomized patients introducing bias in the analysis. What were the reasons for not meeting assay criteria for both assays? What was the source of the tumor material– please describe by primary tumor, metastases, treatment naïve vs post treatment.

4) Line 106 – Suggest adding TR-1 = both subtype and PD-L1 and TR2 = subtype or PD-L1 – this is not clear. In the figures it appears that only 227 had PD-L1 status. Why less than half of the patients?

5) Line 135 “eigengene” – I have to confess that I was not familiar with this term until I looked it up. Suggest explaining what this is.

6) I understand the use of median cutoffs to allow binary analysis of biomarkers. How about looking at these as a continuous variable?

7) Throughout the Results section provide p values for all of the comparisons (only selected ones are provided). When you say “revealed an association between” this implies statistical significance which the reader is left wondering (similar to comment 2 above)

8) 1d – non-bladder sites - please explain what these are (see comment 3 above)

9) Fig 1 e – Please add p values to each panel

10) Fig 2 title should be revised to reflect that d-f assess association with clinical outcomes while a-c reflect associations of biomarkers with subtypes. In the figure description (lines) P value methods are referred to but I don't see these in the figures – please add to each panel (same as comment 10).

Discussion

11) Lines 241 – “High PD-L1 was also associated with high immune signature scores”. This is already noted in the preceding paragraph (line 232-233) and seems out of place here.

Methods

Consider evaluating with the TCGA classifier (Eur Urol PMID:30851984) since this is a major source of the Decipher test and is the largest data set comprising the consensus classifier.

August 16, 2021

Dear Reviewers,

On behalf of all authors and trial investigators, we want to thank you for providing comments from reviewers and editors on our manuscript titled, "*Predictive biomarkers for survival benefit with ramucirumab in urothelial cancer: analysis of the RANGE phase 3 trial*".

We have addressed all comments and provided a detailed response to each of the comments and queries below.

Sincerely,

Michiel S. van der Heijden, M.D., Ph.D.

REVIEWER COMMENTS

Reviewer #1 (Remarks to the Author): with expertise in biostatistics, biomarkers and prostate cancer

van der Heijden and colleagues have undertaken a retrospective correlative study of samples from the RANGE trial (which reported positive for PFS and negative for OS). Both transcriptome (Decipher) and PD-L1 IHC data is generated, and these are used to identify associations between subgroups and response, and between epidemiologic features and molecular ones. The statistical and bioinformatics analysis appears to be poor, with many lacking controls and incorrect analyses -- but the Methods are very terse, and it's possible many analyses are correct and just poorly reported. Changes in the Methods could significantly change the reporting, potentially leading to removal of statements around subtype-specificity and others, which would significantly change the impact of this report.

1. Statistics

* Adjustments for multiple-testing are needed throughout, for example Figure 1a-c gives six separate p-values while figure 1f gives 15 and figure 3a gives 20

Thank you for your comment. We have included Benjamini-Hochberg adjusted significance levels for 5% and 10% false discovery rate (FDR) in several locations, including:

- Supplementary Table 1 (page 46 of the revised manuscript)
- Supplementary Table 2 (page 47 of the revised manuscript)
- Supplementary Table 4 (page 50 of the revised manuscript)

Accordingly, we have added to the Statistical analysis subsection in the Methods section of the manuscript:

- "Due to the post hoc exploratory nature, a two-sided 5% significance level has been used for interpretation purposes. However, BH-adjusted significance levels to maintain 5% and 10% FDR were also calculated." (page 18 of the revised manuscript).

We are of the opinion that in pre-planned, primary analyses of clinical trials, when multiple comparisons are made, all p-values should be adjusted for multiplicity to control familywise, type I error. However, in this instance, these analyses were post hoc and exploratory in nature, hence we sought to balance

maintaining sufficient statistical power to detect a potentially meaningful signal while clarifying the exploratory nature of the analyses and using caution when interpreting results. Our approach of not adjusting for multiple comparisons was also taken by Rothman (1990), in a similar context to ours, and it has the added benefit of leading to fewer errors of interpretation when the data under evaluation are not random numbers but actual observations on nature.

Overall, we believe the approach we use for post hoc analyses in the main text along with additional analysis reported in the supplementary section is appropriate for identifying interesting hypotheses rather than confirming them.

*** CPS discretization threshold was not rationalized**

Thank you for pointing this out. Briefly, we used $CPS \geq 10$ as the threshold in this work to be aligned with the cutoff used in the FDA-approved companion diagnostic for identifying UC patients for treatment with the PD-L1 inhibitor pembrolizumab. In our study, samples were assessed for PD-L1 using an approved, commercially available assay (PD-L1 IHC 22C3 pharmDx). As per your suggestion, we now cite these details in the Methods section, and have added text to clarify the rationale for using $CPS \geq 10$:

“CPS algorithm, as described in the PD-L1 IHC 22C3 pharmDx Interpretation Manual for UC²⁰, assesses PD-L1 expression in both tumor and immune cells. CPS is calculated by the number of PD-L1-stained cells (tumor cells, lymphocytes, macrophages) divided by the total number of viable tumor cells, multiplied by 100. Specimens were categorized as PD-L1 high if $CPS \geq 10$ and PD-L1 low if $CPS < 10$, in line with the threshold described in the PD-L1 IHC 22C3 pharmDx Interpretation Manual for UC and the registration for pembrolizumab in first-line cisplatin ineligible locally advanced or metastatic UC.²⁰” (page 14-15 of the revised manuscript)

In order to further investigate the $CPS \geq 10$ cutoff and address subsequent reviewer concerns about using CPS as an integer rather than a continuous measure, we further investigated CPS using a subpopulation treatment effect pattern plot (STEPP). This plot is now added as Supplementary Fig. 1a.

We have added a sentence to the Results section, stating:

“Examination of the $CPS \geq 10$ cutoff was also supported graphically with a subpopulation treatment effect pattern plot (STEPP), by the observation that the 95% CI of the HR consistently appears below 1 when the subpopulation median CPS was approximately above 10 (Supplementary Fig. 1a).” (page 6 of the revised manuscript)

The corresponding Methods have also been described:

“Continuous measurements for PD-L1 CPS, angiogenesis and immune mean signature scores were examined using the STEPP method³⁰ to determine if the magnitude of the treatment effect changed as a function of the values of these biomarkers. STEPP analyses help determine whether the treatment effect changed for subpopulations with different biomarker values. This is done by dividing the total patient population into overlapping subpopulations based on different biomarker thresholds (known as the sliding window approach), estimating the treatment effect within each subpopulation, and plotting treatment effects against biomarker values. In PD-L1 CPS analysis, 60 patients were included in each window and there were 40 patients in common between two consecutive windows. In analyses of angiogenesis and immune mean signatures, 120 patients were included in each window and there were 60 patients in common between two

consecutive windows. HRs were plotted against median biomarker values in each window.”
(page 18 of the revised manuscript)

Please note, this is also relevant for our response to the next reviewer comment.

* CPS is continuous (or at least integer) and should be analyzed as such throughout

As noted in the above response, our rationale for using a cutoff of $CPS \geq 10$ is supported by the PD-L1 IHC 22C3 pharmDx Interpretation Manual for UC, and text describing this has been added to the revised manuscript.

Furthermore, we believe that since the main interest in these results is understanding the treatment effect, and modeling/reporting treatment effect of ramucirumab, investigating CPS in a continuous fashion may not be intuitive, particularly for interpretation and future use. Thus, we decided a priori to report results based on the $CPS \geq 10$ cutoff. Importantly, $CPS \geq 10$ is an accepted clinical cut-off which corresponds to the registration label of pembrolizumab.

However, to further investigate the continuous CPS and the $CPS \geq 10$ cutoff, we utilized the STEPP method (as described above) showing hazard ratios and 95% confidence intervals, by median PD-L1 CPS (Supplementary Fig. 1a of the revised manuscript).

We have added a sentence clarifying this in the Results section of the revised manuscript:

“Examination of the $CPS \geq 10$ cutoff was also supported graphically with a subpopulation treatment effect pattern plot (STEPP), by the observation that the 95% CI of the HR consistently appears below 1 when the subpopulation median CPS was approximately above 10 (Supplementary Fig. 1a).” (page 6 of the revised manuscript)

* Statistical methods are insufficiently detailed

As per your suggestion, we have added a more detailed description of the Statistical analyses (page 17 and 18 of the revised manuscript). For specific edits made, please see individual responses below, as well as detailed responses to concerns raised by Reviewer 2.

* the repeated $p=2.00e-16$ values likely reflect numerical estimation limits, not true p-values tied with exactly that value

We thank the reviewer for pointing this out. We have revised the manuscript to report such instances as $p < 0.0001$.

* the statistics and reporting of subtypes do not match. The statistics provide a one-way ANOVA (i.e. means are not all equal) while the text discusses "likelihood to be $CPS < 10$ " and "subgroup with the second highest score". These concepts are not statistically equivalent: either restate the text to match the analysis or update the analysis to match the hypothesis being considered.

We agree with the reviewer that these concepts could be explained better. We have done this in the following manner:

In Figure 2a (the relationship between CPS score and molecular subtype), we have made a concise illustration of the continuous distribution of CPS with one-way ANOVA comparison of means. In the Results section, we have added more pertinent description and statistical test (i.e., chi-square/Fisher's test):

"Analysis of the association between molecular subtype and PD-L1 CPS revealed that basal tumor types of UC classification schemes were more likely to be CPS ≥ 10 : 52.2% of Basal and 82.1% of Claudin-Low subtypes using the Decipher GSCv1 classifier²¹, 73.5% of the Basal/Squamous subtype using the ConsensusMIBC classifier¹¹ (Fig. 2a), and 76.9% of the Basal Squamous subtype using TCGA were CPS ≥ 10 (data not shown). In contrast, luminal tumor types of UC classification schemes were more often CPS < 10 . For example, 80.3% of Luminal and 70.4% of Luminal Infiltrated using the Decipher GSCv1 classifier and 91.7% (Luminal Papillary), 68.4% (Luminal Non-Specified), and 52.5% (Luminal Unstable) subtypes using the ConsensusMIBC classifier were CPS < 10 (Fig. 2a). A chi-square test for CPS ≥ 10 versus Decipher GSCv1 subtypes was significant ($p < 0.0001$). A Fisher's exact test for CPS ≥ 10 versus ConsensusMIBC subtypes was also significant ($p = 0.0005$). In addition, Basal and Claudin Low of Decipher GSCv1 subtypes and Basal/Squamous of ConsensusMIBC classification were associated with higher CPS levels when it was investigated as a continuous variable (Fig. 2a). (page 7 of the revised manuscript)

* Similarly, the text discusses which subtypes "benefited most from ramucirumab", but this cannot be determined from the numerical comparison of HRs, this requires a statistical comparison of them (e.g., a likelihood ratio test or (better) an interaction term). Numerous statements suffer from this problem, for example "confirming the greatest ramucirumab treatment benefit is seen in basal type tumors". And of course, multiple testing needs to be appropriately adjusted for in these analyses.

We thank the reviewer for another important suggestion. The evaluation of interactions has been added to all comparisons in either figures or text, as outlined below:

a) We have added a sentence to the Results section stating,

"Interactions between treatment and immune signature scores ($> \text{median}$ vs $\leq \text{median}$) were significant for the T-cell inflamed ($p = 0.04$), activated CD4 T-cell ($p = 0.01$), and memory CD4 T-Cell ($p = 0.04$)." (page 7 of the revised manuscript)

b) We also added a sentence to the Results stating,

"The interaction between treatment and Decipher GSCv1 subtypes (Claudin-Low vs others) is not significant ($p = 0.32$), while the interaction between treatment and ConsensusMIBC subtypes (Basal/Squamous vs others) is marginally insignificant ($p = 0.058$)." (page 8 of the revised manuscript)

Considering the interaction usually requires triple the sample size to be sufficiently powered to detect the significance relative to the marginal effects, we think $p = 0.058$ represents a reasonable signal of true correlation. We hope this provides a rationale to keep our original conclusion that basal subtype benefited most from ramucirumab.

Kindly note that multiple-testing adjustment is not performed for interaction. Please refer to the first comment for the rationale for this.

* Outline the way in which the PH assumption was verified for all Cox models, and specifically state which (if any) models showed violations: it's surprising with this many models that none are reported

We thank the reviewer for this suggestion. Proportional hazard (PH) assumptions have been examined for all Cox models. The assumption is violated for only one model: the unstratified Cox model in the subgroup of CPS ≥ 10 . The footnote of PH assumption violation has been added to Figure 1. Footnotes for all other Cox models have been added to figure legends to state that the PH assumption has not been violated.

* Multivariate modeling rather than repeated univariate modeling is needed in Figure 3 to identify a region effect independent of other covariates

Our apologies if this was not clear in the manuscript. As indicated in the text and figure legend, Figure 3 is indeed based on a multivariate analysis. For clarification (and for review purpose only), mean of angiogenesis/immune signatures are modeled according to the linear regression model below:

$$Y_i = \beta_0 + \beta_1 I\{Gender_i = Male\} + \beta_2 I\{Region_i = East\ Asia\} + \beta_3 Age_i + \beta_4 I\{Histology_i = Pure\ Transitional\ Cell\} + \beta_5 I\{Primary\ Tumor\ Site_i = Bladder\} + \beta_6 I\{Visceral\ Metastases_i = Yes\} + \beta_7 I\{Liver\ Metastases_i = Yes\} + \beta_8 Bellmunt_i + \beta_9 I\{Prior\ Therapy_i = Neoadjuvant\} + \beta_{10} I\{Prior\ Therapy_i = None\} + \varepsilon_i$$

where Y represents mean of angiogenesis signatures in the left panel and immune signatures in the right panel (Figure 3); $\beta_0 \dots \beta_{10}$ are the coefficients and ε_i denotes the random error of patient i .

2. Bioinformatics

* The module eigengene (ME) score is not interpretable in its current form. It appears that a PCA was done on ~7 angiogenesis signatures and separately on ~6 immune signatures, then the loadings of the first PC of each was used as a summary measure. There does not appear to be any QC of the PCA, nor an indication of percent variance explained, nor an analysis of the correlation structure of the scores, and so forth. There is no rationale given for why a PCA is superior to simply taking the median or mean score

Thank you for this critical suggestion. Based on this comment and that of Reviewer #2 (Comment #5), we have replaced module eigengene (ME) score with mean signature score, to summarize the angiogenesis and immune signature pathway sets for ease of interpretation.

Kindly note that the correlation matrices below are provided here for review purposes only. The angiogenesis and immune signature sets each showed significant, positive intercorrelations ($p < 0.001$). The *mean* signature set score was strongly correlated with individual signatures (angiogenesis r^2 0.71–0.92; immune r^2 0.52–0.93).

Angiogenesis mean signature (red box)
 correlations with individual angiogenesis signatures (n=7) *** p<0.001

Immune mean signature (red box)
 correlations with individual immune signatures (n=6) *** p<0.001

Based on results of this Pearson correlation analysis (PCA), the use of mean signature score to represent a single angiogenesis and immune score appears reasonable and use of PCA is not warranted. It is important to note that reporting as a mean signature score does not change our overall results.

We further examined the use of a single mean signature score, for each angiogenesis and immune score, graphically with the STEPP method, (Supplementary Fig. 1b and 1c of the revised manuscript). We added a sentence to the Results section of the manuscript:

“Additionally, to explore the association between ramucirumab benefit and angiogenesis and immune mean signature score in a continuous manner, the STEPP method was utilized (Supplementary Fig. 1b and 1c). The 95% CI of the HR was consistently below 1 when the subpopulation median immune signature score went above 3.88, confirming the finding in dichotomized analysis that high immune signature scores provided OS ramucirumab benefit. However, no clear trend was observed for the angiogenesis mean signature score.” (page 7 of the revised manuscript)

*** Bioinformatics methods are insufficiently detailed**

Our apologies for not being detailed. As per the critique, we have provided more details to give context for the genomics data generation as well as bioinformatics methods. Particularly, in the Pathway signature and analysis subsection of the Methods, we added the following paragraph:

“Gene signature scores were calculated as the mean of the log2 RMA normalized gene expression values for genes comprising the signatures, and the corresponding z-scores were used for heatmaps. Signatures were selected based on postulated related biology and the seven angiogenesis and six immune signatures showed significant, positive intercorrelations within each set (p<0.001). In addition, the average of each signature set showed significant, positive correlations with the individual signatures comprising the set (angiogenesis r² 0.71–0.92, p<0.001; immune r² 0.52–0.93, p<0.001). The signature set average was included for analysis as single angiogenesis and immune score and evaluated relative to results obtained with individual signatures.” (page 16 of the revised manuscript)

* No clear controlling for tumour purity in transcriptome analyses, for example using deconvolution algorithms

Thank you for pointing this out. Tumor purity was addressed as follows:

“A sufficient number of slides (N=10) per tumor sample with 5uM thick tissue sections were provided to Decipher for expression profiling with n=1 (of N=10) slide used for H&E staining to assess tumor cellularity. H&E guided macro-dissection was done to achieve at least 0.5 mm² with a tumor percent macro-dissected area of ≥50% to control for tumor purity.” (page 15 of the revised manuscript)

3. Data Availability

* The DECIPHER data comprises a key component of the value of this study and needs to be made available both to readers and reviewers.

We highly appreciate the interest from Reviewer #1 in data underlying the results presented in this manuscript. We do intend to share anonymized clinical and associated biomarker data including gene expression data presented here with the scientific community. However, this will be done in accordance with Eli Lilly and Company policy to manage potential conflict of interest, potential intellectual property concerns, patient privacy, and related matters. This was perhaps not visibly stated in the prior manuscript version and has been placed prominently in the current version and stated below.

“Eli Lilly and Company will provide access to all individual participant data collected during the trial, after anonymization, except for pharmacokinetic or genetic data. Data are available to request 6 months after the indication studied has been approved in the USA and EU or after primary publication acceptance, whichever is later. No expiration date of data requests is currently set once data have been made available. Access to data will be provided after a proposal has been approved by an independent review committee identified for this purpose and after receipt of a signed data sharing agreement. Data and documents, including the study protocol, statistical analysis plan, clinical study report, and blank or annotated case report forms, will be provided in a secure data sharing environment. For details on submitting a request, see the instructions provided at www.vivli.org.”

Eli Lilly and numerous other pharma companies utilize Vivli’s platform to share clinical data in a secure manner (see <https://vivli.org/members/ourmembers/>). Eli Lilly and Company’s data sharing policies are listed here (see <https://vivli.org/ourmember/lilly/>) along with details on what is typically shared as well as the data request procedure.

“Data requests are initially reviewed by Vivli and Lilly for completeness and other parameters (relating to scope, system-compatibility, and meeting sponsor policies) and are then reviewed by a fully independent review panel in a process that is managed by the Wellcome Trust.” (see full text here: <https://vivli.org/ourmember/lilly/>)

Additionally, given that each data access request requires a formal application and evaluation, it would be extremely challenging for us to create an exception by providing this information without any data sharing agreement in place. This may further complicate the anonymous nature of this review.

A list of Eli Lilly's clinical studies available in Vivli can be found here, (<https://search.vivli.org/?search=Eli%20lilly>) where more than 400 studies have been made available. In summary, we are committed to sharing data with the broader scientific community. However, we would greatly appreciate being allowed to manage sharing underlying data in accordance with Eli Lilly and Company policy.

Reviewer #2 (Remarks to the Author): with expertise in urologic oncology

This manuscript describes a biomarker study embedded in the RANGE study that evaluated Ramucirumab for treatment of patients with platinum refractory bladder cancer. Please consider the following comments:

Introduction

1) Line 77-78 – the use of the term “intrinsic” is not uniformly agreed upon as it relates to these subtypes. Suggest deleting this term.

We thank the reviewer for this suggestion and have deleted the term “intrinsic” and now simply say “...based on “basal-like” and “luminal” cell type classifications”. (page 4 of the revised manuscript)

2) Line 91-93 – I am not aware that a “numeric improvement” is meaningful when the OS endpoint is non-significant. Please rephrase noting that OS was non-significant without reference to “numerical improvement”

As per the suggestion, we have reworded this to:

“Follow-up analyses confirmed the PFS benefit, without a significant improvement in OS in the intent-to-treat (ITT) population (OS 9.4 vs 7.9 months, ramucirumab + docetaxel vs docetaxel only).¹⁹” (page 5 of the revised manuscript)

Results

3) Lines 103-105 – please include the total number of samples submitted to Decipher so we can understand the % that passed QC for both subtype and PD-L1. 227 ends up representing only 43% of the randomized patients introducing bias in the analysis.

We thank the reviewer for pointing out this lack of clarity. We had previously included these details in the Methods section of the submitted manuscript (under Statistical analysis). We agree with the reviewer that readers may benefit from having this information highlighted in Results. We have added this information to the Results:

“The ITT analysis population of the RANGE trial included 530 randomized (1:1) patients. Of these, 462 patient tumor samples were available to submit to Decipher Biosciences for gene expression profiling. Of these samples, 394 met assay criteria for gene expression profiling and 227 samples additionally met PD-L1 IHC assay criteria. Therefore, two cohorts comprised the translational research (TR) populations: TR1 (gene expression profiling and PD-L1 IHC, n=227) and TR2 (gene expression profiling, n=394).” (page 5 of the revised manuscript)

With respect to assessment of bias in the analysis, we show that baseline demographics and disease characteristics of the two translational research (TR) populations, TR1/TR2, are representative of the overall RANGE ITT population, with the exception of a lower proportion of East Asian patients in the TR1 (n=227) population (Table 1). We have ensured a description of this is included in the Results section of the manuscript:

“Baseline demographics/disease characteristics of TR1/TR2 populations were representative of the overall RANGE ITT population with the exception of a lower proportion of East Asian patients in the TR1 (PD-L1) population (3.1% in the TR1 population vs 20.8% in RANGE ITT and 18.8% in TR2) (Table 1).” (page 5 of the revised manuscript)

What were the reasons for not meeting assay criteria for both assays?

For the PD-L1 IHC assay, sample slides older than 6 months after sectioning were out of the assay window and therefore not included. We have added a sentence to the Sample collection and immunohistochemistry subsection of the Methods:

“If more than 6 months had passed since sectioning, those sample slides were considered out of the stability window and were excluded from analyses for purposes of this exploratory biomarker study.” (page 14 of the revised manuscript)

The Decipher assay criteria for gene expression profiling were described in the Methods section:

“Tumor samples from 462 patients in the RANGE phase 3 trial were submitted to Decipher Biosciences (previously GenomeDx) for tumor sample gene expression profiling and UC molecular subtyping. A sufficient number of slides (N=10) per tumor sample with 5µM thick tissue sections were provided to Decipher for expression profiling with n=1 (of N=10) slide used for H&E staining to assess tumor cellularity. H&E guided macro-dissection was done to achieve at least 0.5 mm² with a tumor percent macro-dissected area of ≥50% to control for tumor purity. Whole transcriptome analysis was performed on formalin-fixed, paraffin-embedded tumor tissue with GeneChip® Human Exon 1.0 ST Array (Affymetrix) in a Clinical Laboratory Improvement Amendments-certified laboratory. Expression data were generated on 410/462 samples with 32 samples failing for low tumor content, one failing due to low RNA quantity, and 19 failing at cDNA preparation stage. CEL files were processed with the Transcriptome Analysis Console version 4.0 software (Thermo Fisher Scientific, Santa Clara CA). Of 410 samples, 394 passed microarray quality control (16 failed) and were included in these analyses. Samples were normalized using Robust Multi-Array (RMA)³³.” (page 15 of the revised manuscript)

What was the source of the tumor material— please describe by primary tumor, metastases, treatment naïve vs post treatment.

A majority of tumor samples came from diagnostic biopsy and were thus predominantly treatment naïve. Of the n=394 samples used for gene expression profiling, n=278 were from primary tumor, n=77 were from metastatic tumor, and the remaining n=39 did not report any such detail.

4) Line 106 – Suggest adding TR-1 = both subtype and PD-L1 and TR2 = subtype or PD-L1 – this is not clear. In the figures it appears that only 227 had PD-L1 status. Why less than half of the patients?

As per the suggestion, we have edited this paragraph for clarity:

“Therefore, two cohorts comprised the translational research (TR) populations: TR1 (gene expression profiling and PD-L1 IHC, n=227) and TR2 (gene expression profiling, n=394).” (page 5 of the revised manuscript)

With respect to baseline demographics and disease characteristics of the two translational research (TR) populations, TR1/TR2 are representative of the overall RANGE ITT population, with the exception of a lower proportion of East Asian patients in the TR1 (n=227) population (Table 1). We have ensured a description of this is in the Results section of the manuscript:

“Baseline demographics/disease characteristics of TR1/TR2 populations were representative of the overall RANGE ITT population with the exception of a lower proportion of East Asian patients in the TR1 (PD-L1) population (3.1% in the TR1 population vs 20.8% in RANGE ITT and 18.8% in TR2) (Table 1).” (page 5 of the revised manuscript)

The main reason for this is due to PD-L1 IHC assay criteria: slides had to be <6 months from time of sectioning. Many of the samples were only available as slides, not tissue blocks (especially those from patients from East Asia) and were beyond the 6-month window

The Sample collection and immunohistochemistry of the Methods section now states:

“If more than 6 months had passed since sectioning, those sample slides were out of the stability window and were excluded from analyses for the purposes of this exploratory biomarker study.” (page 14 of the revised manuscript)

Additionally, in the Discussion, we mention the smaller sample size of the TR1 (n=227) population as a limitation of our study:

“Given the assay requirements of PD-L1 and the age of the tumor samples (in some cases, only slides were available), only a subset of the RANGE tumors could be assessed for PD-L1, leading to a smaller patient population in TR1 as well as fewer East Asian samples with PD-L1 data. Furthermore, because East Asian sites enrolled early, a disproportionate number of samples from that region were past the 6-month window for evaluation of PD-L1.” (pages 12-13 of the revised manuscript)

5) Line 135 “eigengene” – I have to confess that I was not familiar with this term until I looked it up. Suggest explaining what this is.

Based on this feedback and that of Reviewer #1, we have removed summarization using module eigengene, and have instead used mean signature score for ease of interpretation, without any change in overall results.

Angiogenesis and immune signature sets each showed significant, positive intercorrelations ($p < 0.001$). The mean signature set score was strongly correlated with individual signatures (angiogenesis r^2 0.71–0.92; immune r^2 0.52–0.93). Please see the correlation matrices below, which we present for reviewer purposes only.

The signature set average was included for treatment effect analysis as single signature set metric and evaluated relative to individual signature results.

As noted above in response to Reviewer #1, we further examined the use of a single mean signature score, for each angiogenesis and immune score, graphically utilizing the STEPP method, (Supplementary Fig. 1b and 1c of the revised manuscript). We also added a sentence to the Results section of the manuscript:

“Additionally, to explore the association between ramucirumab benefit and angiogenesis and immune mean signature score in a continuous manner, the STEPP method was utilized (Supplementary Fig. 1b and 1c). The 95% CI of the HR was consistently below 1 when the subpopulation median immune signature score went above 3.88, confirming the finding in dichotomized analysis that high immune signature scores provided OS ramucirumab benefit. However, no clear trend was observed for the angiogenesis mean signature score.” (page 7 of the revised manuscript)

6) I understand the use of median cutoffs to allow binary analysis of biomarkers. How about looking at these as a continuous variable?

As per the suggestion, we did explore how the magnitude of treatment effect changed as a function of biomarker in a continuous fashion using the STEPP method. The findings are consistent with the binary analyses. We reported the STEPP method of three primary biomarkers in the supplementary material, including PD-L1 CPS, angiogenesis, and immune mean signature scores. We have added a sentence clarifying this in the Results section of the revised manuscript:

“Examination of the CPS ≥ 10 cutoff was also supported graphically with a subpopulation treatment effect pattern plot (STEPP), by the observation that the 95% CI of the HR consistently appears below 1 when the subpopulation median CPS was approximately above 10 (Supplementary Fig. 1a).” (page 6 of the revised manuscript)

And

“Additionally, to explore the association between ramucirumab benefit and angiogenesis and immune mean signature score in a continuous manner, the STEPP method was utilized (Supplementary Fig. 1b and 1c). The 95% CI of the HR was consistently below 1 when the subpopulation median immune signature score went above 3.88, confirming the finding in dichotomized analysis that high immune signature scores provided OS ramucirumab benefit. However, no clear trend was observed for the angiogenesis mean signature score.” (page 7 of the revised manuscript)

7) Throughout the Results section provide p values for all of the comparisons (only selected ones are provided). When you say “revealed an association between” this implies statistical significance which the reader is left wondering (similar to comment 2 above)

All p-values and other statistical information were included in the Figure legends. However as suggested and to help clarify, we have added p-values directly to the Figure panels, where appropriate, and cited these within the Results.

8) 1d – non-bladder sites - please explain what these are (see comment 3 above)

“Non-bladder” primary tumor site refers to tumors originating in the renal pelvis, ureter, or urethra. We have added a footnote to Table 1 to indicate this. In addition, we have added a footnote indicating that “Other” primary tumor site refers to tumors with more than one primary site (page 29-31 of the revised manuscript).

9) Fig 1 e – Please add p values to each panel

As suggested, p-values have been added directly to each panel, where applicable.

10) Fig 2 title should be revised to reflect that d-f assess association with clinical outcomes while a-c reflect associations of biomarkers with subtypes. In the figure description (lines) P value methods are referred to but I don't see these in the figures – please add to each panel (same as comment 10).

As per the suggestion, we have edited the title of Figure 2: *“Molecular subtype association with PD-L1 status, immune/angiogenesis signatures, and clinical outcome”*.

Also, as suggested, p-values have been added directly to the Figure 2 panels.

Discussion

11) Lines 241 – “High PD-L1 was also associated with high immune signature scores”. This is already noted in the preceding paragraph (line 232-233) and seems out of place here.

Thank you, we have deleted this repetitive sentence (page 11 of the revised manuscript).

Methods

Consider evaluating with the TCGA classifier (Eur Urol PMID:30851984) since this is a major source of the Decipher test and is the largest data set comprising the consensus classifier.

Thank you for this suggestion. We applied the TCGA classifier as per [de Jong JJ, Liu Y, Robertson AG, Seiler R, Groeneveld CS, van der Heijden MS, et al. Genome Med. 2019;11(1):60], which was based on the published TCGA 2017 subtypes. Additionally, heatmaps have been updated with TCGA 2017 classification in Supplementary Figure 3A and B. The TCGA 2017 scheme is well aligned with the ConsensuMIBC classification scheme, as shown in the new Supplementary Table 3. In particular, the Basal/Squamous subtype of both classification schemes show considerable overlap.

We have also added KM plots for overall survival using the TCGA 2017 classification (Supplementary Figure 2 of the revised manuscript), and a sentence to the results stating

“The association of OS with Basal/Squamous subtype was also consistent when using a TCGA 2017 classification (Supplementary Fig. 2).” (page 8 of the revised manuscript)

Reviewers' Comments:

Reviewer #1:

Remarks to the Author:

The authors have appropriately addressed several of my prior concerns, but others remain for further consideration.

1. Statistics

a) Multiple-Testing

Unfortunately the authors have elected to only partially address this concern. Multiple testing correction is not optional: false discovery rates should be reported throughout, especially given that this is a discovery study. This reflects the long-established standard practice in this field, and indeed it would be deceptive to readers to report anything else (or to mix FDR and non-FDR results). Some of the current "hits" will be lost because there is a lower level of statistical evidence, and that is an accurate reporting of the results.

b) PH Assumptions

Please outline in the Methods how these were assessed. The current text only states "Proportional hazards assumptions were checked and reported (lines 492-493)"

c) Multivariate Modeling

I apologize for not recognizing that Figure 3 was indeed multivariate in the prior version of the manuscript! Please add the provided equation to the Methods, it greatly clarifies the manuscript and reduces the imprecision of interpretation.

2. Bioinformatics

a) ME Scores

The authors have substantially improved this analysis. The correlation matrices provided for review purposes are key for interpretation of the (strong!) robustness of the results by readers. Please add them as supplementary figures, and ensure their generation is documented in the Methods.

b) Tumor Purity

The authors have not yet controlled for tumor purity using standard deconvolution methods and/or by explicitly controlling for estimated purity in statistical modeling. The broad influence of tumor purity on transcriptome associations is well-known, and a 50% threshold means that up to half the total signal (and potentially a much higher proportion for individual genes) derives from non-tumor cells. This proportion will vary from tumor to tumor, introducing an uncontrolled for variance (and potentially bias) into all analyses.

3. Data Availability

a) Unfortunately the current data statement still indicates "Eli Lilly and Company will provide access to all individual participant data collected during the trial, after anonymization, except for pharmacokinetic or genetic data." Thus the data availability statement currently appears to exclude the DECIPHER data itself.

b) From a review perspective, the current situation leaves the reviewer unable to actually review the data itself and compromises the review quality

c) From a reader perspective, there is no clear mechanism for data-sharing

d) The Vivli link (<https://vivli.org/ourmember/lilly/>) explicitly states "access to data may be declined... where there is a potential conflict of interest or an actual or potential competitive risk". This is concerning: a sponsor may decide a biomarker researcher is a "potential competitive risk" and there is no wording or safeguards to prevent serious limitations to use being imposed post-publication.

e) No clear DOI or other link is provided to the VIVLI page, and the reviewer was unable to identify it on the VIVLI site. A search of other trials (e.g. <https://search.vivli.org/studyDetails/5da7a153-1217-4669-adab-578727ede2b7>) did not provide any ability to determine what data was available for any given study.

f) As noted in the original review, the DECIPHER data comprises a key component of the study value, and needs to be made easily and clearly available.

Reviewer #2:

Remarks to the Author:

The authors have done a meticulous job of addressing my concerns. Regarding the last comment on Methods, the referenced classifier from Genome Medicine is not the same as the one described in the Eur Urol reference - deJong et al developed their own single patient classifier using the RNAseq data from the TCGA. However, for the purposes of the present manuscript it should suffice.

Reviewer #1 Comments

The authors have appropriately addressed several of my prior concerns, but others remain for further consideration.

1. Statistics

a) Multiple-Testing

Unfortunately the authors have elected to only partially address this concern. Multiple testing correction is not optional: false discovery rates should be reported throughout, especially given that this is a discovery study. This reflects the long-established standard practice in this field, and indeed it would be deceptive to readers to report anything else (or to mix FDR and non-FDR results). Some of the current "hits" will be lost because there is a lower level of statistical evidence, and that is an accurate reporting of the results.

Thank you for providing us another opportunity to address this concern. We have made substantial revisions in this round regarding the concern of multiple testing.

Specifically, corrections were made if multiple testing of the same hypothesis was conducted, i.e. the hypothesis of superior treatment effect for ramucirumab, using Benjamini-Hochberg-adjusted p-values. We have included BH-adjusted p-values in the Results section for all results reporting on the hypothesis of superior treatment effect of ramucirumab. We have also added details to the Methods section, which read:

“For hypotheses with a single comparison, a two-sided 5% significance level has been used. For the hypothesis of superior treatment effect from ramucirumab, multiple comparisons were conducted in specified subgroups. BH-adjusted p-values were calculated and reported to account for multiple comparisons. Due to the post hoc, exploratory nature of the analyses, BH-adjusted p-values below 20% have been highlighted throughout the manuscript for interpretation purposes.” (page 18 of revised manuscript)

Finally, Supplementary Table 1 has been added which reports all subgroups that were investigated for the hypothesis of superior treatment effect for ramucirumab. For each subgroup, we report the stratified p-value, the rank of the stratified p-value, and the BH-adjusted p-value.

b) PH Assumptions

Please outline in the Methods how these were assessed. The current text only states "Proportional hazards assumptions were checked and reported (lines 492-493)

Proportional hazard assumptions were checked using the score test based on weighted residual proposed in the following paper: Grambsch, P.M. & Therneau, T.M. Proportional hazards tests and diagnostics based on weighted residuals. *Biometrika* 81, 515-526 (1994).

A sentence with citation to this paper has been clarified in the Methods section, which now reads:

“Proportional hazard assumptions were checked using the score test based on weighted residual⁴⁹ and reported for all Cox regression models.” (page 17 of revised manuscript)

c) Multivariate Modeling

I apologize for not recognizing that Figure 3 was indeed multivariate in the prior version of the

manuscript! Please add the provided equation to the Methods, it greatly clarifies the manuscript and reduces the imprecision of interpretation.

Per your request, we have added the following model to the Methods section of the revised manuscript. This section now reads:

“The association between clinical covariates and mean signature score was assessed using a multivariable linear regression model for both angiogenesis and immune signature, respectively. The coefficients of linear regression models were estimated for all clinical covariates according to the following regression model and reported:

$$Y_i = \beta_0 + \beta_1 I\{Gender_i = Male\} + \beta_2 I\{Region_i = East Asia\} + \beta_3 Age_i + \beta_4 I\{Histology_i = Pure Transitional Cell\} + \beta_5 I\{Primary Tumor Site_i = Bladder\} + \beta_6 I\{Visceral Metastases_i = Yes\} + \beta_7 I\{Liver Metastases_i = Yes\} + \beta_8 Bellmunt_i + \beta_9 I\{Prior Therapy_i = Neoadjuvant\} + \beta_{10} I\{Prior Therapy_i = None\} + \varepsilon_i$$

where Y_i represents angiogenesis mean signature score in the left panel and immune mean signature score in the right panel of patient i (Figure 3); $\beta_0 \dots \beta_{10}$ are the coefficients; $I\{\}$ represents the indicator function for categorical variables; and ε_i denotes the random error of patient i .” (pages 17-18 of revised manuscript)

2. Bioinformatics

a) ME Scores

The authors have substantially improved this analysis. The correlation matrices provided for review purposes are key for interpretation of the (strong!) robustness of the results by readers. Please add them as supplementary figures, and ensure their generation is documented in the Methods.

Thank you for the suggestion. We have added the correlation matrices as Supplementary Figure 5 and documented it in the Methods section.

b) Tumor Purity

The authors have not yet controlled for tumor purity using standard deconvolution methods and/or by explicitly controlling for estimated purity in statistical modeling. The broad influence of tumor purity on transcriptome associations is well-known, and a 50% threshold means that up to half the total signal (and potentially a much higher proportion for individual genes) derives from non-tumor cells. This proportion will vary from tumor to tumor, introducing an uncontrolled for variance (and potentially bias) into all analyses.

Reviewer #1 raises important points about the use of deconvolution methods in tumor profiling studies.

We agree that solid tumors specimens will comprise not only tumor cells but also several non-malignant yet biologically relevant cell types in their microenvironment such as immune cells, endothelial cells, and stromal cells including cancer-associated fibroblasts. The interaction between tumor and its non-tumor microenvironment plays a critical role in tumor cell biology.

The macro-dissection approach used to enrich for tumor purity in this study is a standard pathology method for enriching neoplastic content upstream of molecular characterization (Ascierto, P.A., et al. *J. Mol. Diagn.* 21, 756-767 [2019]). It should be noted that prior studies, such as TCGA, selected samples based on requirements of >100 mg of surgical tissue and >60% tumor nuclei. Given that we had

biopsies, we intended to achieve $\geq 50\%$ tumor purity, a reasonable criterion for clinical studies, and one that is on par with sample sets from which the molecular classifications of muscle invasive bladder cancer (MIBC) were derived without the application of deconvolution methods (Kamoun, A., et al. *Eur. Urol.* 77, 420-433 [2020]; Seiler, R., et al. *Eur. Urol.* 72, 544-554 [2017]).

While a variety of deconvolution methods are available for segregating tumor samples into component cell types and fractions, we did not adopt this approach for several reasons.

Firstly, the suggested adjustment could artificially impact assessment of the biological variance of interest: the main mode of action of an anti-angiogenic agent (ramucirumab) is on the stromal compartments of the tumor microenvironment. More specifically, tumor intrinsic endothelial cells are the main drug target and therapeutic activity on other stromal compartments, such as immune modulation, could play a role here as well.

Secondly, in a comprehensive study looking at this topic, authors assessed impact of tumor purity on genomic analyses in TCGA (Aran, D., et al. *Nat. Commun.* 6, 8971 [2015]) and demonstrated that replicates from the same patient tended to show similar purity using the same method. Importantly, based on breast cancer subtypes, authors of this study noted that differences in subtype purity might be genuine and intrinsic characteristics of the subtypes themselves.

Thirdly, in line with the above-mentioned reasons, and specifically in the context of MIBC, stromal and immune compartments have a potentially meaningful association within the molecular subtypes of interest as noted in Kamoun et al 2020, Supplementary Figure 3 using ABSOLUTE algorithm and Efstathiou, J.A., et al. *Eur. Urol.* 76:69-70 (2019), which aligns with the mode of action of our study drug. Another related example is noted in a recent paper describing the molecular determinants of response to PD-L1 agents across tumor types including bladder cancer (Banchereau, R., et al. *Nature Comm.* 12:3969 [2021]) where a $>20\%$ tumor purity criteria was used. Multiple non-epithelial microenvironmental cell types were described that allowed for understanding the overall microenvironmental contribution to treatment benefit in a retrospective setting.

Lastly, as noted in Supp Figures 2, 3, and 5 from Aran D, et al. 2015, the purity estimates can vary dramatically depending on the method deployed e.g., H&E, DNA (ABSOLUTE), stroma/immune cells (ESTIMATE), Methylation (LUMP). This indicates that work remains to be done before we can utilize these approaches as a standard methodology in relevant circumstances.

In summary, the thoughtful comment from Reviewer #1 regarding the composition of tumor profiled is valid for a study focusing on gene expression of the neoplastic cells within a tumor. However, for the reasons outlined above, we think that the approach employed is the most biologically and potentially clinically relevant variance of interest in this experimental context. In response to this criticism, we have acknowledged this potential limitation in the Discussion section, which now reads:

“Additionally, heterogeneity of the tumor tissue sampled by the biopsy and further by the macrodissection process may impact the molecular subtype classification.” (page 12 of revised manuscript)

We sincerely hope that the reviewer will find this satisfactory.

3. Data Availability

a) Unfortunately the current data statement still indicates "Eli Lilly and Company will provide access to all individual participant data collected during the trial, after anonymization, except for pharmacokinetic or genetic data." Thus the data availability statement currently appears to exclude the Decipher data itself.

b) From a review perspective, the current situation leaves the reviewer unable to actually review the data itself and compromises the review quality

c) From a reader perspective, there is no clear mechanism for data-sharing

For comments a) to c) above, we would like to clarify that Decipher gene expression data were generated using Affymetrix microarray technology and as such would not be classified as genetic data. Very briefly, genetic data here would refer to germline sequencing or SNP data that can potentially identify an individual. As noted in point f) below, Decipher expression data are now being made available.

d) The Vivli link (<https://vivli.org/ourmember/lilly/>) explicitly states "access to data may be declined... where there is a potential conflict of interest or an actual or potential competitive risk". This is concerning: a sponsor may decide a biomarker researcher is a "potential competitive risk" and there is no wording or safeguards to prevent serious limitations to use being imposed post-publication.

We would like to clarify that it is only in exceptional circumstances as noted at the link above, where data requests may be declined, "*In exceptional circumstances, access to data may be declined by the sponsor, for example, where there is a potential conflict of interest or an actual or potential competitive risk*".

e) No clear DOI or other link is provided to the VIVLI page, and the reviewer was unable to identify it on the VIVLI site. A search of other trials (e.g., <https://search.vivli.org/studyDetails/5da7a153-1217-4669-adab-578727ede2b7>) did not provide any ability to determine what data was available for any given study.

We apologize for the inconvenience this may have caused. Anonymized patient level data and related information from RANGE can be requested through vivli (<https://search.vivli.org/?search=NCT02426125>). Alternately, this can be done by searching for ramucirumab and looking for the relevant study (<https://search.vivli.org/?search=Ramucirumab>).

f) As noted in the original review, the Decipher data comprises a key component of the study value, and needs to be made easily and clearly available.

As per the request by the reviewer, we have added the Decipher data as a Supplementary dataset. Accordingly, we have amended the data availability statement to indicate that the Decipher data are available as a Supplementary data file. The amended portion of the data availability statement reads:

"The Decipher gene expression profiling data matrix has been made available as a Supplementary dataset." (page 18 of revised manuscript)

Reviewer #2 (Remarks to the Author):

The authors have done a meticulous job of addressing my concerns. Regarding the last comment on Methods, the referenced classifier from Genome Medicine is not the same as the one described in the Eur Urol reference - deJong et al developed their own single patient classifier using the RNAseq data from the TCGA. However, for the purposes of the present manuscript it should suffice.

Thank you for your careful review of our manuscript. We appreciate your time and attention.

References

- Aran, D., Sirota, M., & Butte, A.J. Systematic pan-cancer analysis of tumour purity. *Nat. Commun.* **6**, 8971 (2015).
- Ascierto, P.A., et al. Preanalytic variables and tissue stewardship for reliable next-generation sequencing (NGS) clinical analysis. *J. Mol. Diagn.* **21**, 756-767 (2019).
- Banchereau, R., et al. Molecular determinants of response to PD-L1 blockade across tumor types. *Nat. Commun.* **12**, 3969 (2021).
- Efstathiou, J.A., et al. Impact of immune and stromal infiltration on outcomes following bladder-sparing trimodality therapy for muscle-invasive bladder cancer. *Eur. Urol.* **76**, 69-70 (2019).
- Grambsch, P.M. & Therneau, T.M. Proportional hazards tests and diagnostics based on weighted residuals. *Biometrika* **81**, 515-526 (1994).
- Kamoun, A., et al. A consensus molecular classification of muscle-invasive bladder cancer. *Eur. Urol.* **77**, 420-433 (2020).
- Seiler, R., et al. Impact of molecular subtypes in muscle-invasive bladder cancer on predicting response and survival after neoadjuvant chemotherapy. *Eur. Urol.* **72**, 544-554 (2017).

Reviewers' Comments:

Reviewer #3:

Remarks to the Author:

The manuscript has been subject to 2 previous rounds of reviewing and the authors have adequately addressed previous reviewers' comments, with the detailed report attached below. However, there remains a key concern that the association between biomarkers of interest and clinical outcomes were assessed using univariable Cox proportional hazard models only (Fig 1/2). The authors presented a multivariable model in sTable 4 but it included only multiple candidate biomarkers and their interactions with treatment arms. In fact, the limited availability of samples in each individual analysis and the introduction of new biomarkers for stratification have made the experimental and standard arms no longer randomised. In this case, multivariable analysis adjusting for clinical characteristics will be needed to accurately evaluate biomarker-associated treatment benefit from ramucirumab.

Minor updates needed:

1. The names of columns/rows of the correlation matrix in sFig. 5 are too small to read.
2. Please make sure the term multivariate and multivariable were used consistently throughout the manuscript.
3. It will be easier to follow if one more column for p-values can be added in Table 1 to indicate the difference between ITT/TR1/TR2.

Other than these, the paper is in a mature shape and will be ready to publish.

Addressing Reviewer 1's comments

1. Multiple comparisons and proportional hazard assumptions have been addressed to a satisfactory level.
2. The correlation matrix has been added as requested, but please make a clearer label on the name of each column/row in sFig 5, especially the ones in the angiogenesis set.

The argument that anti-angiogenesis treatment would have an impact not only on tumor but also on surrounding microenvironment is convincing and adding the discussion point of tumor sample purity in the manuscript is fair.

3. Data availability statement has been improved. Clinical trial data at an individual level has long been shared on a by-request basis. It is important that the authors now make available the gene expression data.

REVIEWER COMMENTS

Reviewer #3 (Remarks to the Author):

The manuscript has been subject to 2 previous rounds of reviewing and the authors have adequately addressed previous reviewers' comments, with the detailed report attached below. However, there remains a key concern that the association between biomarkers of interest and clinical outcomes were assessed using univariable Cox proportional hazard models only (Fig 1/2). The authors presented a multivariable model in sTable 4 but it included only multiple candidate biomarkers and their interactions with treatment arms. In fact, the limited availability of samples in each individual analysis and the introduction of new biomarkers for stratification have made the experimental and standard arms no longer randomised. In this case, multivariable analysis adjusting for clinical characteristics will be needed to accurately evaluate biomarker-associated treatment benefit from ramucirumab.

We thank the reviewer for raising this important point. We agree that the randomization/balance of baseline clinical factors could be jeopardized by the limited samples in each biomarker subgroup analysis. In order to address that we reported stratified HR/p-values for all Cox models in this manuscript. As noted in Petrylak, D.P., et al. *Lancet Oncol.* 21, 105-120 [2020], stratification was based on geographical region, baseline ECOG PS, and visceral metastases, which were known prognostic factors. We chose stratified Cox model to avoid potential overfitting in analysis with limited sample size. When the prognostic effect of clinical factors are known and are not of interest in a particular analysis, a stratified Cox model has fewer coefficients to estimate and therefore could provide higher efficiency than a multivariable Cox model.

We have revised the Methods section to make this clearer. The revision reads:

“When reporting results of stratified Cox regression, clinical trial stratification factors including geographical region, baseline ECOG PS, and visceral metastases were used to adjust for possible imbalance across treatment arms within each biomarker subgroup or molecular subtype.”

Additionally, we have ensured that all Figure legends and Table titles clearly describe that models were stratified, where needed.

Minor updates needed:

1. The names of columns/rows of the correlation matrix in sFig. 5 are too small to read.

We have revised the correlation matrix to include larger, more legible column/row labels for the mean and individual signatures. Thank you for this comment which greatly improves the readability of these figures.

2. Please make sure the term multivariate and multivariable were used consistently throughout the manuscript.

Thank you for pointing this discrepancy out. We have revised to multivariable throughout.

3. It will be easier to follow if one more column for p-values can be added in Table 1 to indicate the difference between ITT/TR1/TR2.

Please note, these comparisons were meant to be descriptive and not predictive. Furthermore, both TR populations are subsets of the ITT population, and the TR1 population is a subset of the TR2 population. We added a statement to the Statistical analysis sub-section of the Methods to clarify this fact. This addition reads:

“Of note, both TR populations are a subset of the ITT population, and TR1 is a subset of TR2.”

Other than these, the paper is in a mature shape and will be ready to publish.

Addressing Reviewer 1's comments

1. Multiple comparisons and proportional hazard assumptions have been addressed to a satisfactory level.

2. The correlation matrix has been added as requested, but please make a clearer label on the name of each column/row in sFig 5, especially the ones in the angiogenesis set.

The argument that anti-angiogenesis treatment would have an impact not only on tumor but also on surrounding microenvironment is convincing and adding the discussion point of tumor sample purity in the manuscript is fair.

3. Data availability statement has been improved. Clinical trial data at an individual level has long been shared on a by-request basis. It is important that the authors now make available the gene expression data.

Thank you for your detailed review of our manuscript. We appreciate your time and comments.

Reviewers' Comments:

Reviewer #3:

Remarks to the Author:

The authors' explanation on the use of the stratified Cox model has adequately addressed my primary concern of the analysis used in this paper. The paper is now in a satisfactory format ready for publication.

REVIEWERS' COMMENTS

Reviewer #3 (Remarks to the Author):

The authors' explanation on the use of the stratified Cox model has adequately addressed my primary concern of the analysis used in this paper. The paper is now in a satisfactory format ready for publication.

Thank you for your detailed review of our manuscript. We appreciate your time and comments.